# Unitary Evolution and Elements of Reality in Consecutive Quantum Measurements

**DOI:** 10.3390/e24070877

**Published:** 2022-06-26

**Authors:** Dmitri Sokolovski

**Affiliations:** 1Departmento de Química-Física, Universidad del País Vasco, UPV/EHU, 48940 Leioa, Spain; dgsokol15@gmail.com; 2IKERBASQUE, Basque Foundation for Science, Plaza Euskadi 5, 48009 Bilbao, Spain

**Keywords:** quantum measurements, unitary evolutions, quantum elements of reality, 03.65.Ta, 03.65.AA, 03.65.UD

## Abstract

Probabilities of the outcomes of consecutive quantum measurements can be obtained by construction probability amplitudes, thus implying the unitary evolution of the measured system, broken each time a measurement is made. In practice, the experimenter needs to know all past outcomes at the end of the experiment, and that requires the presence of probes carrying the corresponding records. With this in mind, we consider two different ways to extend the description of a quantum system beyond what is actually measured and recorded. One is to look for quantities whose values can be ascertained without altering the existing probabilities. Such “elements of reality” can be found, yet they suffer from the same drawback as their EPR counterparts. The probes designed to measure non-commuting operators frustrate each other if set up to work jointly, so no simultaneous values of such quantities can be established consistently. The other possibility is to investigate the system’s response to weekly coupled probes. Such weak probes are shown either to reduce to a small fraction the number of cases where the corresponding values are still accurately measured, or lead only to the evaluation of the system’s probability amplitudes, or their combinations. It is difficult, we conclude, to see in quantum mechanics anything other than a formalism for predicting the likelihoods of the recorded outcomes of actually performed observations.

## 1. Introduction

In [1], Feynman gave a brief yet surprisingly thorough description of quantum behaviour. Quantum systems are intrinsically stochastic, calculation of probabilities must rely on complex valued probability amplitudes, and it is unlikely that one will be able to get a further insight into the mechanism behind the formalism. One may ask two separate questions about the view expressed in [1]. Firstly, is it consistent? There have been recent suggestions [2] that quantum mechanics may be self-contradictory, and that its flaws can be detected from within the theory, i.e., by considering certain thought experiments. In [3], we have, argued that the proposed “contradictions” are easily resolved if Feynman’s description is adopted. Secondly, one can ask if the rules can be explained further? There have been proposals of “new physics” based on such concepts as time symmetry, weak measurements, and weak values (see [4,5,6], and Refs. therein). Recently, we have shown the weak values to be but Feynman’s probability amplitudes, or their combinations [7,8]. The ensuing paradoxes occur if the amplitudes are used inappropriately, e.g., as a proof of the system’s presence at a particular location [9], a practice known for quite some time to be unwise (see [10] and Ch. 6 pp. 144–145 of [11]). It is probably fair to say that Feynman’s conclusions have not been successfully challenged to date, and we will continue to rely on them in what follows.

The approach of [1] is particularly convenient for describing situations where several measurements are made one after another on the same quantum system. Such consecutive or sequential measurements have been studied by various authors over a number of years [12,13,14,15,16], and we will continue to study them here. The simplest case involves just two observations, of which the first prepares the measured system in a known state, and the second yields the value of the measured quantity “in that state.” Adding intermediate measurements between these two significantly changes the situation, as it brings in a new type of interference the measurements can now destroy. Below we will discuss two particular issues which arise in the analysis of such sequential measurements. One is the break down of the unitary evolution of the measured system, which occurs each time a measurement is made. Another is the possibility of extending the description of the system beyond what is actually being measured. This can be done, e.g., by looking for “elements of reality”, i.e., the properties or values which can be ascertained without changing anything else in the system’s evolution. This can also be done by studying a system’s response to weekly coupled inaccurate measuring devices. It is not our purpose here to dispute the findings made by the authors using alternative approaches (see, for example, [5]). Rather we we want to see how the above issues can be addressed in conventional quantum mechanics, as presented in [1].

The rest of the paper is organised as follows. In Section 2, we recall the basic rules and discuss the broken unitary evolution of the measured system, In Section 3, we note that, in order to be able to gather the statistics, the experimenter would need the records of the previous outcomes. The system’s broken evolution can then be traded for an unbroken unitary evolution of a composite {system + the probes which carry the records}. In Section 4, we discuss two different (and indeed well known) types of the probes. In Section 5, we discuss the quantities whose additional measurements would not alter the likelihoods of all other outcomes. However, like their EPR counterparts, these “elements of reality” cannot be observed simultaneously. In Section 6, we illustrate this on a simple two-level example. In Section 7, we look at what would happen in an attempt to measure two of such quantities jointly. Section 8 asks if something new can be learnt about the system by minimising the perturbation incurred by the probes. Section 9 contains a summary of our conclusions.

## 2. Feynman’s Rules of Quantum Motion: Broken Unitary Evolutions

Consider a system (S) with which the theory associates *N*-dimensional Hilbert space $H_{S}$. The L+1 quantities ${\hat{Q}}^{\ell }$ to be measured at the times $t_{0}\ldots <t_{\ell }\ldots <t_{L}$ are represented by Hermitian operators ${\hat{Q}}^{\ell }$ acting in $H_{S}$, each with $M_{\ell }\leq N$ distinct real valued eigenvalues $Q_{m_{\ell }}^{\ell }$
(1)$$\begin{array}{c} {\hat{Q}}^{\ell }=\sum_{m_{\ell }=1}^{M_{\ell }}Q_{m_{\ell }}^{\ell }\sum_{n_{\ell }}^{N}\Delta (Q_{m_{\ell }}^{\ell }-\langle q_{n_{\ell }}^{\ell }|{\hat{Q}}^{\ell }|q_{n_{\ell }}^{\ell }\rangle )|q_{n_{\ell }}^{\ell }\rangle \langle q_{n_{\ell }}^{\ell }|\equiv \sum_{m_{\ell }=1}^{M_{\ell }}Q_{m_{\ell }}^{\ell }{\hat{\pi }}_{m_{\ell }}^{\ell }, \end{array}$$
where $|q_{n_{\ell }}^{\ell }\rangle$, ($\langle q_{n_{\ell }}^{\ell }|q_{n_{\ell }^{\prime }}^{\ell }\rangle =\delta_{n_{\ell }n_{\ell }^{\prime }}$, $n_{\ell }=1,\ldots N$) are the measurement bases, Δ(X−Y)=1 if X=Y, and 0 otherwise, and ${\hat{\pi }}_{m_{\ell }}$ is the projector onto the eigen-subspace, corresponding to an eigenvalue $Q_{m_{\ell }}^{\ell }$. The first operator ${\hat{Q}}^{0}$ is assumed to have only non-degenerate eigenvalues, i.e., ${\hat{Q}}^{0}=\sum_{n_{0}=1}^{N}Q_{n_{0}}^{0}\left|q_{n_{0}}^{0}\rangle \langle q_{n_{0}}^{0}\right|$. This is needed to initialise the system, in order to proceed with the calculation.

The possible outcomes of the experiment are, therefore, the sequences of the observed values $Q_{m_{L}}^{L}..Q_{n_{0}}^{0}$, and one wishes to predict the probabilities (frequencies) with which a particular *real* path $\left\{Q_{m_{L}}^{L}\ldots \leftarrow Q_{m_{\ell }}^{\ell }\ldots \leftarrow Q_{n_{0}}^{0}\right\}$ would occur after many trials. Following [1], one can obtain these obtained by constructing first the system’s *virtual* paths $\left\{q_{n_{L}}^{L}\ldots \leftarrow q_{n_{\ell }}^{\ell }\ldots \leftarrow q_{n_{0}}^{0}\right\}$, connecting the corresponding states in $H_{S}$, and ascribing to each path a probability amplitude (we use ℏ=1)
(2)$$\begin{array}{c} A\left(q_{n_{L}}^{L}\ldots \leftarrow q_{n_{\ell }}^{\ell }\ldots \leftarrow q_{n_{0}}^{0}\right)=\prod_{\ell =0}^{L-1}\langle q_{n_{\ell +1}}^{\ell +1}\left|{\hat{U}}_{S}\left(t_{\ell +1},t_{\ell }\right)\right|q_{n_{\ell }}^{\ell }\rangle , \end{array}$$
where ${\hat{U}}_{S}\left(t_{\ell +1},t_{\ell }\right)=\exp\left[-i\int_{t_{\ell }}^{t_{\ell +1}}{\hat{H}}_{S}\left(t^{\prime }\right)dt^{\prime }\right]$ is the system’s evolution operator (the time ordered product is assumed if the system’s hamiltonian operators ${\hat{H}}_{S}\left(t^{\prime }\right)$ do not commute at different times, $[{\hat{H}}_{S}\left(t^{\prime }\right),{\hat{H}}_{S}\left(t^{\prime \prime }\right)]\neq 0$). We will assume that all Hermitian operators ${\hat{Q}}^{\ell }={\left({\hat{Q}}^{\ell }\right)}^{†}$ can be measured in this way. We will also allow for all unitary evolutions, ${\hat{U}}_{S}^{†}\left(t_{\ell +1},t_{\ell }\right)={\hat{U}}_{S}^{-1}\left(t_{\ell +1},t_{\ell }\right)$.

Combining the virtual paths according to the degeneracies of the intermediate eigenvalues $Q_{m_{\ell }}^{\ell }$, 1≤ℓ≤L−1, yields elementary paths, endowed with both the amplitudes
(3)$$\begin{array}{c} A(q_{n_{L}}^{L}\ldots \leftarrow Q_{m_{\ell }}^{\ell }\ldots \leftarrow q_{n_{0}}^{0})=\;\;\;\;\;\;\\ \sum_{n_{1}\ldots n_{L-1}=1}^{N}\prod_{\ell =1}^{L-1}\Delta (Q_{m_{\ell }}^{\ell }-\langle q_{n_{\ell }}|{\hat{Q}}^{\ell }\left|q_{n_{\ell }}\rangle \right)A\left(q_{n_{L}}^{L}\ldots \leftarrow q_{n_{\ell }}^{\ell }\ldots \leftarrow q_{n_{0}}^{0}\right).\; \end{array}$$
and the probabilities,
(4)$$\begin{array}{c} p\left(q_{n_{L}}^{L}\ldots \leftarrow Q_{m_{\ell }}^{\ell }\ldots \leftarrow q_{n_{0}}^{0}\right)={\left|A\left(q_{n_{L}}^{L}\ldots \leftarrow Q_{m_{\ell }}^{\ell }\ldots \leftarrow q_{n_{0}}^{0}\right)\right|}^{2} \end{array}$$
We note that the amplitudes in Equation (Equation 3) depend only on the projectors ${\hat{\pi }}_{m_{\ell }}^{\ell }$ in Equation (Equation 2), and not on the corresponding eigenvalues $Q_{m_{\ell }}^{\ell }$. To stress this, we are able to write
(5)$$\begin{array}{c} A\left(q_{n_{L}}^{L}\ldots \leftarrow Q_{m_{\ell }}^{\ell }\ldots \leftarrow q_{n_{0}}^{0}\right)=A\left(q_{n_{L}}^{L}\ldots \leftarrow {\hat{\pi }}_{m_{\ell }}^{\ell }\ldots \leftarrow q_{n_{0}}^{0}\right). \end{array}$$
Finally, summing $p(q_{n_{L}}^{L}\ldots \leftarrow Q_{m_{\ell }}^{\ell }\ldots \leftarrow q_{n_{0}}^{0})$ over the degeneracies of the last operator ${\hat{Q}}^{L}$, yields the desired probabilities for the real paths,
(6)$$\begin{array}{c} P\left(Q_{m_{L}}^{L}\ldots \leftarrow Q_{m_{\ell }}^{\ell }\ldots \leftarrow Q_{n_{0}}^{0}\right)=\sum_{n_{L}}^{N}\Delta (Q_{m_{L}}^{L}-\langle q_{n_{L}}^{L}|{\hat{Q}}^{L}\left|q_{n_{L}}^{L}\rangle \right)p\left(q_{n_{L}}^{L}\ldots \leftarrow {\hat{\pi }}_{m_{\ell }}^{\ell }\ldots \leftarrow q_{n_{0}}^{0}\right). \end{array}$$
Note that there is no interference between the paths leading to different (i.e., orthogonal) final states $|q_{n_{L}}^{L}\rangle$, even if they correspond to the same eigenvalue $Q_{m_{L}}^{L}$ [1]. This is necessary, since an additional (L+2)-nd measurement of an operator ${\hat{Q}}^{L+1}=\sum_{n_{L}=1}^{N}Q_{n_{L+1}}^{L+1}\left|q_{n_{L+1}}^{L}\rangle \langle q_{n_{L+1}}^{L}\right|$ immediately after $t=t_{L}$ would destroy any interference between the paths ending in different $|q_{n_{L}}^{L}\rangle$s at $t=t_{L}$. Since future measurements are not supposed to alter the results already obtained, one never adds the amplitudes for the *final* orthogonal states [1]. Note that the same argument cannot be repeated for the past measurements at $t_{\ell }<t_{L}$.

It may be convenient to cast Equation (Equation 6) in an equivalent form,
(7)$$\begin{array}{c} P\left(Q_{m_{L}}^{L}\ldots \leftarrow Q_{m_{\ell }}^{\ell }\ldots \leftarrow q_{n_{0}}^{0}\right)=\langle \Phi \left({\hat{\pi }}_{m_{L}}^{L}\ldots \leftarrow {\hat{\pi }}_{m_{\ell }}^{\ell }\ldots \leftarrow q_{n_{0}}^{0}\right|\Phi \left({\hat{\pi }}_{m_{L}}^{L}\ldots \leftarrow {\hat{\pi }}_{m_{\ell }}^{\ell }\ldots \leftarrow q_{n_{0}}^{0}\right)\rangle ,\; \end{array}$$
where
(8)$$\begin{array}{c} |\Phi ({\hat{\pi }}_{m_{L}}^{L}\ldots \leftarrow {\hat{\pi }}_{m_{\ell }}^{\ell }\ldots \leftarrow q_{n_{0}}^{0}\rangle =\prod_{\ell =1}^{L}{\hat{\pi }}_{m_{\ell }}^{\ell }\left(t_{\ell },t_{0}\right)|q_{n_{0}}^{0}\rangle ,\\ {\hat{\pi }}_{m_{\ell }}^{\ell }\left(t_{\ell },t_{0}\right)\equiv {\hat{U}}_{S}^{-1}\left(t_{\ell },t_{0}\right){\hat{\pi }}_{m_{\ell }}^{\ell }{\hat{U}}_{S}\left(t_{\ell },t_{0}\right),\; \end{array}$$
and a unitary evolution of the initial state $|q_{n_{0}}^{0}\rangle$ with the system’s evolution operator ${\hat{U}}_{s}$ is seen to be interrupted at each $t=t_{\ell }$. check that the probabilities in Equation (Equation 6) sum, as they should, to unity.

It is worth bearing in mind the Uncertainty Principle which, we recall, states that [1] “*one cannot design equipment in any way to determine which of two alternatives is taken, without, at the same time, destroying the pattern of interference*”. In particular, this means that if two or more virtual paths in Equation (Equation 2) are allowed to interfere, it must be absolutely impossible to find out which one was followed by the system. Moreover, one will not even be able to say that, in a given trial, one of them was followed, while the others were not (see [10] and Ch. 6 pp. 144–145 of [11]).

With the basic rules laid out, and an example given in Figure 1, we will turn to practical realisations of an experiment involving several consecutive measurements of the kind just described.

## 3. The Need for Records: Unbroken Unitary Evolutions

In an experiment, described in Section 2, there are $N\times M_{1}\times M_{2}\ldots \ldots \times M_{L}$ possible sequences of observed outcomes. At the end of each trial, the experimenter identifies the real path followed by the system, $path=\{Q_{m_{L}}^{L}\ldots \leftarrow Q_{m_{\ell }}^{\ell }\ldots \leftarrow Q_{n_{0}}^{0}\}$, and increases by 1 the count in the corresponding part of their inventory, K(path)→K(path)+1. After K>>1 trials, the ratios K(path)/K will approach the probabilities in Equation (Equation 6), from which all the quantities of interest, such as averages or correlations, can be obtained later.

There is one practical point. In order to identify the path, an Observer must have readable records of *all past outcomes* once the experiment is finished, i.e., just after $t=t_{L}$. There are two reasons for that. Firstly, quantum systems are rarely visible to the naked eye, so something accessible to the experimenter’s senses is clearly needed. Secondly, and more importantly, the condition of the system changes throughout the process [cf. Equation (Equation 8)], and its final state simply cannot provide all necessary information. In other words, one requires *L* probes which copy the system’s state at $t=t_{\ell }$, ℓ=0,1,…L and retain this information till the end of the trial. It is easy to see what such probes must do. The experiment begins by coupling the first probe to a previously unobserved system at $t=t_{0}$. To proceed with the calculation, we may assume that just $t_{0}$ the initial state of a composite system+probes is
(9)$$\begin{array}{c} |\Psi_{S+Probes}\left(0\right)\rangle =|q_{n_{0}}^{0}\rangle |D^{0}\left(n_{0}\right)\rangle \ldots |D^{\ell }\left(0\right)\rangle \ldots |D^{L}\left(0\right)\rangle \equiv \left|q_{n_{0}}^{0}\rangle \right|\Psi_{Probes}\left(0\right)\rangle \end{array}$$
where $|D^{\ell }\left(0\right)\rangle$ is the initial state of the *ℓ*-th probe which, if found changed into $|D^{\ell }\left(m_{\ell }\right)\rangle$, $\langle D^{\ell }\left(m_{\ell }\right)|D^{\ell }\left(m_{\ell }^{\prime }\right)\rangle =\delta_{m_{\ell }m_{\ell }^{\prime }}$, would tell the experimenter that the outcome at $t=t_{\ell }$ was $Q_{m_{\ell }}^{\ell }$. Note that the first probe $D^{0}$ has already been coupled to a previously unobserved system and produced a reading $n_{0}$, thus preparing the system in a state $|q_{n_{0}}^{0}\rangle$.

The composite would undergo unitary evolution with an (yet unknown) evolution operator ${\hat{U}}_{S+Probes}\left(t_{L},t_{0}\right)$. The rules of the previous section still apply, albeit in a larger Hilbert space, and with only two (L=1) measurements, of which the first one prepares the entire composite in the state (Equation 9). For simplicity, we let the last operator have non-degenerate eigenvalues, $M_{L}=N$. By (Equation 6), the probability to have an outcome $Q_{n_{L}}^{L}\ldots \leftarrow Q_{m_{\ell }}^{\ell }\ldots \leftarrow Q_{n_{0}}^{0}$ is
(10)$$\begin{array}{c} \tilde{P}\left(Q_{n_{L}}^{L}\ldots \leftarrow Q_{m_{\ell }}^{\ell }\ldots \leftarrow Q_{n_{0}}^{0}\right)=\;\;\;\;\;\;\;\;\;\;\;\;\\ \sum_{n_{L}^{\prime }=1}^{N}{\left|\langle q_{n_{L}^{\prime }}^{L}|\langle \Psi_{Probes}\left(n_{L},\ldots m_{\ell }\ldots n_{0}\right)\left|{\hat{U}}_{S+Probes}\left(t_{L},t_{0}\right)\right|\Psi_{S+Probes}\left(0\right)\rangle \right|}^{2} \end{array}$$
where
(11)$$\begin{array}{c} |\Psi_{Probes}\left(n_{L},\ldots m_{\ell }\ldots n_{0}\right)\rangle =|D^{L}\left(n_{L}\right)\rangle \prod_{\ell =1}^{L-1}\left|D^{\ell }\left(m_{\ell }\right)\rangle \right|D^{0}\left(n_{0}\right)\rangle \end{array}$$
We want the probabilities in Equation (Equation 10) (the ones the experimenter measures) and the probabilities in Equation (Equation 6) (the ones the theory predicts) to agree. Consider again the scenarios $\left\{q_{n_{L}}^{L}\ldots \leftarrow Q_{n_{\ell }}^{\ell }\ldots \leftarrow q_{n_{0}}^{0}\right\}$ in Equation (Equation 3). In the absence of the probes they lead to the same final state, $|q_{n_{L}}^{L}\rangle$, interfere, and cannot be told apart, according to the Uncertainty Principle. If we could use the probes to turn these scenarios into exclusive alternatives [1], e.g., by directing them to different (orthogonal) final states in the larger Hilbert space, Equation (Equation 6) for the system subjected to L+1 measurements would follow. In other words, we will be able to trade a broken evolution in a smaller space $H_{S}$ [cf. Equation (Equation 8)] for an uninterrupted unitary evolution in a larger Hilbert space $H_{S+Probes}$. For this we need an evolution operator ${\hat{U}}_{S+Probes}\left(t_{L},t_{0}\right)$ such that
(12)$$\begin{array}{c} \langle q_{n_{L}^{\prime }}^{L}\left|{\hat{U}}_{S+Probes}\left(t_{L},t_{0}\right)\right|\Psi_{S+Probes}\left(0\right)\rangle =\delta_{n_{L}^{\prime }n_{L}}\times \;\;\;\;\;\\ \sum_{m_{1}\ldots .m_{L-1}=1}^{M_{1}\ldots M_{L-1}}A_{S}\left(q_{n_{L}}^{L}\ldots \leftarrow Q_{m_{\ell }}^{\ell }\ldots \leftarrow q_{n_{0}}^{0}\right)|\Psi_{Probes}\left(n_{L},\ldots m_{\ell }\ldots n_{0}\right)\rangle , \end{array}$$
where the orthogonal states $|\Psi_{Probes}\left(n_{L},\ldots m_{\ell }\ldots n_{0}\right)\rangle$ play the role of “tags”, by which previously interfering paths $\left\{q_{n_{L}}^{L}\ldots \leftarrow Q_{m_{\ell }}^{\ell }\ldots \leftarrow q_{n_{0}}^{0}\right\}$ can now be distinguished (see Figure 2).

For the reader worried about the collapse of the wave function we note that the same probabilities can be obtained in two different ways. Either the evolution of the wave function *of the system only* is broken every time an instantaneous measurement is made, as happens in Equation (Equation 8), or the evolution *of the system + the probes* continues until the end of the experiment as in Equation (Equation 12).

Finally, we note the difference between producing all *L* records, but not using or having no access to some of them, and not producing some of the records at all. There is also a possibility of *destroying*, say, the *ℓ*-th record by making a later measurement on a composite {the system+ the *ℓ*-th probe} [17,18]. In this case, the composite becomes the new measured system, and the rules of the previous section still apply.

## 4. Two Kinds of Probes

We note next that it does not really matter for the theory how exactly the records are produced, as long as the interference between the virtual paths is destroyed, and Equation (Equation 12) is satisfied. The states $|D^{\ell }\rangle$ in Equation (Equation 9) may equally refer to devices, to the Observer or the Observers’ memories [17,19], or to the notes the Observers have made in the course of a trail, [18]. We will assume for simplicity that the probes have no own dynamics, and retain their states after having interacted with the measured system,
(13)$$\begin{array}{c} {\hat{H}}_{S+Probes}={\hat{H}}_{S}+{\hat{H}}_{int},\;{\hat{H}}_{Probes}=0. \end{array}$$
Several interactions which have the desired effect are, in fact, well known, and we will discuss them next. There are at least two types of probes consistent with Equation (Equation 12). They require different treatments, and we will consider them separately.

### 4.1. Discrete Gates

For the *ℓ*-th probe consider a register of $M_{\ell }$ two-level sub-systems, each prepared in its lower states $|1_{m_{\ell }}\rangle$
(14)$$\begin{array}{c} |D^{\ell }\left(0\right)\rangle =\prod_{m_{\ell }=1}^{M_{\ell }}|1_{m_{\ell }}\rangle . \end{array}$$
The probe, designed to measure a quantity ${\hat{Q}}^{\ell }=\sum_{m_{\ell }=1}^{M_{\ell }}Q_{m_{\ell }}^{\ell }{\hat{\pi }}_{m_{\ell }}^{\ell }$, is coupled to the system via
(15)$$\begin{array}{c} {\hat{H}}_{int}^{\ell }=-\left(\pi /2\right)\sum_{m_{\ell }=1}^{M_{\ell }}{\hat{\pi }}_{m_{\ell }}^{\ell }{\hat{\sigma }}_{x}^{\ell }\left(m_{\ell }\right)\delta \left(t-t_{\ell }\right), \end{array}$$
where ${\hat{\sigma }}_{x}^{\ell }\left(m_{\ell }\right)$ is the Pauli matrix, which acts on the $m_{\ell }$-th sub-system in the usual way, ${\hat{\sigma }}_{x}^{\ell }\left(m_{\ell }\right)\left|1_{m_{\ell }}\rangle =\right|2_{m_{\ell }}\rangle$. Since the individual terms in Equation (Equation 15) commute, the evolution operator of the {System+ℓ−thProbe} over a short interval $\left[t_{\ell }-\epsilon ,t_{\ell }+\epsilon \right]$, ϵ→0 is
(16)$$\begin{array}{c} {\hat{U}}_{int}^{\ell }\left(t_{\ell }\right)=\exp\left[i\left(\pi /2\right)\sum_{m_{\ell }=1}^{M_{\ell }}{\hat{\pi }}_{m_{\ell }}^{\ell }{\hat{\sigma }}_{x}^{\ell }\left(m_{\ell }\right)\right]=i\sum_{m_{\ell }=1}^{M_{\ell }}{\hat{\pi }}_{m_{\ell }}^{\ell }\sigma_{x}\left(m_{\ell }\right). \end{array}$$
The probe entangles with the system in the required way,
(17)$$\begin{array}{c} {\hat{U}}_{int}^{\ell }\left(t_{\ell }\right)|\psi_{S}\rangle |D^{\ell }\left(0\right)\rangle =i\sum_{m_{\ell }=1}^{M_{\ell }}{\hat{\pi }}_{m_{\ell }}^{\ell }\left|\psi_{S}\rangle \right|D^{\ell }\left(m_{\ell }\right)\rangle , \end{array}$$
where in $|D^{\ell }\left(m_{\ell }\right)\rangle$ is obtained from $|D^{\ell }\left(0\right)\rangle$ by flipping the state of the $m_{\ell }$-th sub-system,
(18)$$\begin{array}{c} |D^{\ell }\left(m_{\ell }\right)\rangle \equiv \left|2_{m_{\ell }}\rangle \prod_{k_{\ell }\neq m_{\ell }}\right|1_{k_{\ell }}\rangle . \end{array}$$
We note that, whatever the state $|\psi_{S}\rangle$, one of the subsystems will change its condition (the system will be found somwhere). We note also that in each trial only one subsystem will be affected (the system is never found simultaneously in two or more places). The full evolution operator is, therefore, given by
(19)$$\begin{array}{c} {\hat{U}}_{S+Probes}\left(t_{L},t_{0}\right)={\hat{U}}_{int}^{\ell }\left(t_{L}\right)\prod_{\ell =0}^{L-1}{\hat{U}}_{S}\left(t_{\ell +1},t_{\ell }\right){\hat{U}}_{int}^{\ell }\left(t_{\ell }\right) \end{array}$$
where, as before, we we assumed $M_{0}=M_{L}=N$, ${\hat{\pi }}_{n_{0}}=\left|q_{n_{0}}\rangle \langle q_{n_{0}}\right|$, and ${\hat{\pi }}_{n_{L}}=\left|q_{n_{L}}\rangle \langle q_{n_{L}}\right|$. The experimenter prepares all probes in the states (Equation 14) and, once the experiment is finished, only needs to check which sub-system of $D^{\ell }$, say, the $m_{\ell }$-th, has changed its state. This will tell him/her that the value of ${\hat{Q}}^{\ell }$ at $t=t_{\ell }$ was $Q_{m_{\ell }}^{\ell }$. As a simple example, Figure 3 shows an outcome of five measurements made on a four-state system. There the first probe, capable of distinguishing between all four states prepares the system in a state $|q_{n_{0}\rangle }^{0}$. The second probe cannot tell apart the third and the second states, so ${\hat{\pi }}_{1}^{1}=\left|q_{1}^{1}\rangle \langle q_{1}^{1}\right|$, ${\hat{\pi }}_{2}^{1}=\left|q_{2}^{1}\rangle \langle q_{2}^{1}\right|$, and ${\hat{\pi }}_{3}^{1}=|q_{3}^{1}\rangle \langle q_{3}^{1}\left|+\right|q_{4}^{1}\rangle \langle q_{4}^{1}|$, and so on. The sequence of the measured valued obtained by inspecting the probes at $t>t_{4}$ is $Q_{4}^{4}\leftarrow Q_{2}^{3}\leftarrow Q_{2}^{2}\leftarrow Q_{3}^{1}\leftarrow Q_{1}^{0}$. After many trials the sequence will be observed with a probability $P\left(Q_{4}^{4}\leftarrow Q_{2}^{3}\leftarrow Q_{2}^{2}\leftarrow Q_{3}^{1}\leftarrow Q_{1}^{0}\right)={\left|A_{S}\left(q_{4}^{4}\leftarrow {\hat{\pi }}_{2}^{3}\leftarrow {\hat{\pi }}_{2}^{2}\leftarrow {\hat{\pi }}_{3}^{1}\leftarrow q_{1}^{0}\right)\right|}^{2}$ [cf. Equation (Equation 5)]

### 4.2. Von Neumann’s Pointers

In classical mechanics one can measure the value of a dynamical variable Q(x,p) at $t_{0}$ by coupling the system to a “pointer”, a heavy one-dimensional particle with position *f* and momentum λ. The full Hamiltonian is given by $H_{S}\left(x,p\right)+\lambda Q\left(x,p\right)\delta \left(t-t_{0}\right)$, and at $t=t_{0}$ the pointer is rapidly displaced by $\delta f=Q(x\left(t_{0}\right),p\left(t_{0}\right))$, which providies the desired reading. What happens to the system, depends on the pointer’s momentum, which remains unchanged by the interaction. If λ=0, the system continues on its way unperturbed. If λ≠0, the system experiences a sudden kick, whereby its position and momentum are changed by $\Delta x=\lambda \partial_{p}Q\left(x\left(t_{0}\right),p\left(t_{0}\right)\right)$ and $\Delta p=\lambda \partial_{x}Q\left(x\left(t_{0}\right),p\left(t_{0}\right)\right)$, respectively.

The quantum version of the pointer [20] employs a coupling ${\hat{H}}_{int}=g\left(t\right)\hat{\lambda }\hat{Q}$, where $\hat{\ell }$, $\langle f^{\prime }\left|\hat{\lambda }\right|f\rangle =-i\delta \left(f-f^{\prime }\right)\partial_{f}$ is the pointer’s momentum operator, and $\hat{Q}=\sum_{m}Q_{m}{\hat{\pi }}_{m}$ is the (system’s) operator to be measured. The function g(t)=1/τ can be chosen constant for the duration of the measurement τ, $t_{0}\leq t\leq t_{0}+\tau$, and zero otherwise. It tends to a Dirac delta $\delta (t-t_{0})$ for an instantaneous (impulsive) measurement, where τ→0. For a system whose state $|\psi_{S}\rangle$ lies in the eigen sub-space of a projector ${\hat{\pi }}_{m}$, the action of ${\hat{H}}_{int}$ results in a spatial shift of the pointer’s initial state |G(0)〉 by $Q_{m}$,
(20)$$\begin{array}{c} \exp\left(-i{\hat{H}}_{int}\tau \right)|\psi_{s}\rangle |G\left(0\right)\rangle =|\psi_{s}\rangle \int G\left(f-Q_{m}\right)\left|f\rangle df\equiv \right|\psi_{s}\rangle |G\left(m\right)\rangle \end{array}$$
With L+1 pointers employed to measure L+1 quantities ${\hat{Q}}^{\ell }$, the initial state of the composite can be chosen to be [cf. Equation (Equation 9)]
(21)$$\begin{array}{c} |\Psi_{S+Pointers}\left(0\right)\rangle =|q_{n_{0}}^{0}\rangle |G^{0}\left(n_{0}\right)\rangle \ldots \left|G^{\ell }\left(0\right)\rangle \ldots \right|G^{L}\left(0\right)\rangle , \end{array}$$
where the initial pointer states can be, e.g., identical Gaussians of a width Δf, all centred at the origin,
(22)$$\begin{array}{c} \langle f_{\ell }|G^{\ell }\left(0\right)\rangle =C\exp\left(-f_{\ell }^{2}/\Delta f^{2}\right)\equiv G^{\ell }\left(f_{\ell }\right),\;C_{1}={\left[2/\pi \Delta f^{2}\right]}^{1/4}, \end{array}$$
except for the first probe, where we would need a narrow Gaussian, $|G^{0}\left(f_{0}\right){|}^{2}\to \delta \left(f\right)$ in order to prepare the system in $|q_{n_{0}}^{0}\rangle$. If all couplings are instantaneous, for the amplitude in Equation (Equation 3) [with L=2, and $|q_{n_{L}}^{L}\rangle$ replaced by a state of the composite, $|q_{n_{L}}^{L}\rangle ⊗|\bar{f}\rangle$, $|\bar{f}\rangle \equiv |f_{0}\rangle ⊗\left|f_{1}\rangle \ldots ⊗\right|f_{L}\rangle$] one finds
(23)$$\begin{array}{c} A_{S+Pointers}\left(\bar{f},q_{n_{L}^{\prime }}^{L}\leftarrow \Psi_{0}\right)=\;\;\;\;\;\;\;\;\;\\ \langle \bar{f}|\langle q_{n_{L}^{\prime }}^{L}\left|{\hat{U}}_{S+Pointers}\left(t_{L},t_{0}\right)\right|\Psi_{S+Pointers}\left(0\right)\rangle =\delta_{n_{L}^{\prime }n_{L}}G^{L}\left(f_{L}-Q_{n_{L}}\right)\times \;\\ \sum_{m_{1}\ldots m_{L-1}=1}^{M_{1}\ldots M_{L-1}}\prod_{\ell =1}^{L-1}G^{\ell }\left(f_{\ell }-Q_{m_{L}}\right)G^{0}\left(f_{0}-Q_{n_{0}}\right)A_{S}\left(q_{n_{L}}^{L}\ldots \leftarrow Q_{m_{\ell }}^{\ell }\ldots \leftarrow q_{n_{0}}^{0}\right)\;\; \end{array}$$
where, again, $A_{S}\left(q_{n_{L}}^{L}\ldots \leftarrow Q_{n_{\ell }}^{\ell }\ldots \leftarrow Q_{n_{0}}^{0}\right)$ is the system’s amplitude (Equation 3).

If one wants their measurements to be accurate, the pointers need to be set to zero with as little uncertainty as possible (see. Figure 4). This uncertainty is determined by the Gaussian’s width Δf, and sending it to zero we have [since $G^{\ell }{\left(f_{\ell }\right)}^{2}\to \delta \left(f_{\ell }\right)$ and $G^{\ell }\left(f_{\ell }\right)G^{\ell }\left(f_{\ell }-X\right)=0$ for any X≠0]
(24)$$\begin{array}{c} P_{Pointers}\left(f_{L}\ldots f_{1},f_{0}\right)\equiv \sum_{n_{L}}{\left|A_{S+Pointers}\left(\bar{f},q_{n_{L}}^{L}\leftarrow \Psi_{0}\right)\right|}^{2}=\\ \sum_{m_{1}\ldots m_{L}=1}^{M_{1}\ldots M_{L}}\prod_{\ell =0}^{L}\delta \left(f_{\ell }-Q_{m_{\ell }}^{\ell }\right)P_{S}\left(Q_{m_{L}}^{L}\ldots \leftarrow Q_{m_{\ell }}^{\ell }\ldots \leftarrow Q_{n_{0}}^{0}\right), \end{array}$$
where $P_{S}\left(Q_{m_{L}}^{L}\ldots \leftarrow Q_{m_{\ell }}^{\ell }\ldots \leftarrow Q_{n_{0}}^{0}\right)$ is the probability, computed with the help of Equation (Equation 7).

Equation (Equation 24) is the desired result, which deserves a brief discussion. In each trial, the pointers’ readings may take only discrete values $Q_{m_{\ell }}^{\ell }$, and the observed sequences occur with the probabilities, predicted for the system by Feynman’s rules of Section 2. However, unlike in the classical case, this information comes at the cost of perturbing the system’s evolution. Indeed, writing $G^{\ell }\left(0\right)=\int G^{\ell }\left(\lambda_{\ell }\right)\exp\left(i\lambda_{\ell }f_{\ell }\right)d\lambda_{\ell }$ and proceeding as before, one obtains terms like $\prod_{\ell =1}^{L-1}\exp\left(-i\lambda_{\ell }{\hat{Q}}^{\ell }\right){\hat{U}}_{S}\left(t_{\ell },t_{\ell -1}\right)$, where $\exp(-i\lambda_{\ell }{\hat{Q}}^{\ell })$ represents the “kick”, produced on the system by the *ℓ*-th pointer at $t_{\ell }$. As in the classical case, we can get rid of the kick by ensuring that the pointer’s momentum $\lambda_{\ell }$ is approximately zero. However, Heisenberg’s uncertainty principle (see, e.g., [1]) will make the uncertainty in the initial pointer position very large. Accuracy and perturbation go hand in hand, and the measured values do not “pre-exist” measurements but are produced in the course of it [21]. Notably, one can still predict the probabilities by not mentioning the pointers at all and analyzing instead an isolated system, whose unitary evolution is broken each time the coupling takes place.

Secondly, and importantly, von Neumann pointers have many states, and only a few of them are actually used. This suggests that the pointers and the probes of the previous section could, in principle, be replaced by much more complex devices, with only a few states of their vast Hilbert spaces coming into play. For example, there is nothing in quantum theory that forbids using printers, which print the observed values on a piece of paper. If an experiment that measures spin’s component is set properly, the machine will print only “up” or “down” with the predicted frequencies, and would never digress into French romantic poetry.

## 5. The Past of a Quantum System: Elements of Reality

The stock of an experiment described so far, is taken just after the last measurement at $t=t_{L}$. This is the “present” moment, the times $t_{0},\ldots t_{L}$ are relegated to the “past”, and the “future” is yet unknown. Possible pasts are defined by the choice of the measured quantities ${\hat{Q}}^{\ell }$, and of the times $t_{\ell }$ at which the impulsive measurements are performed. The $N\times M_{1}\times M_{2}\ldots ..\times M_{L}$ possible outcomes $\left\{Q_{m_{L}}^{L}\ldots \leftarrow Q_{m_{\ell }}^{\ell }\ldots \leftarrow Q_{n_{0}}^{0}\right\}$ occur with probabilities $P(Q_{m_{L}}^{L}\ldots \leftarrow Q_{m_{\ell }}^{\ell }\ldots \leftarrow Q_{n_{0}}^{0})$ which the theory aims to predict. There are clearly gaps in the description of the system between successive measurements at $t_{\ell }$ and $t_{\ell +1}$. One way to fill them (without adding new measurements, which would change the problem) is to look for quantities whose values can be ascertained at some $t_{\ell }<t^{\prime }<t_{\ell +1}$ without altering the existing probabilities $P(Q_{m_{L}}^{L}\ldots .\leftarrow Q_{m_{\ell +1}}^{\ell +1}\leftarrow Q_{m_{\ell }}^{\ell }\ldots \leftarrow Q_{n_{0}}^{0})$. Or, to put it slightly differently, to ask what can be measured without destroying the interference between the virtual paths [cf. (Equation 2)] which contribute to the amplitudes $A_{S}\left(Q_{m_{L}}^{L}\ldots \leftarrow Q_{m_{\ell +1}}^{\ell +1}\leftarrow Q_{m_{\ell }}^{\ell }\ldots \leftarrow Q_{n_{0}}^{0}\right)$ in Equation (Equation 3).

There is a well-known analogy. EPR-like scenarios [22] are often used to question the manner in which quantum theory describes the physical world. In a nutshell, the argument goes as follows. Alice and Bob, at two separate locations, share an entangled pair of spins. Alice can ascertain that Bob’s spin has any desired direction, while apparently unable to influence it due to the restrictions imposed by special relativity. Hence, all possible values of the spin’s projections can exist simultaneously, i.e., be in some sense *real*. If quantum mechanics insists that different projections cannot have well-defined values at the same time, it must be incomplete. We are not interested here in the details of this important ongoing discussion (for an overview see [22] and Refs. therein), or the implications relativity theory may have for elementary quantum mechanics [23]. Rather we want to make use of the Criterion of Reality (CR) used by the authors of [24] to determine what should be considered “real”. This criterion reads: *“If, without in any way disturbing a system, we can predict with certainty (i.e., with probability equal to unity) the value of a physical quantity, then there exists an element of reality corresponding to that quantity.”* [24].

Consider again an experiment in which L+1 measurements are made on the system at $t=t_{\ell }$, ℓ=0,1…L, while the system’s condition at some $t^{\prime }$ between, say, $t_{\ell }$ and $t_{\ell +1}$ remains unknown. To fill this gap, one may use the CR criterion just cited, and look for any information about the system, which can be obtained without altering the existing statistical ensemble. Thus, one needs a variable ${\hat{Q}}^{\prime }$ whose measurement at $t_{\ell }<t^{\prime }<t_{\ell +1}$ results is
(25)$$\begin{array}{c} A_{S}\left(q_{n_{L}}^{L}\ldots \leftarrow Q_{m_{\ell +1}}^{\ell }\leftarrow Q_{m^{\prime }}^{\prime }\leftarrow Q_{m_{\ell }}^{\ell }\ldots \leftarrow q_{n_{0}}^{0}\right)=A_{S}\left(q_{n_{L}}^{L}\ldots \leftarrow Q_{m_{\ell +1}}^{\ell +1}\leftarrow Q_{m_{\ell }}^{\ell }\ldots \leftarrow q_{n_{0}}^{0}\right). \end{array}$$
There are at least two kinds of quantities that satisfy this condition. To the first kind belong operators of the type
(26)$$\begin{array}{c} {\hat{Q}}^{-}\left(t^{\prime }\right)=\sum_{m_{-}}Q_{m_{-}}^{-}{\hat{\pi }}_{m_{-}}^{-}\equiv {\hat{U}}_{S}\left(t^{\prime },t_{\ell }\right){\hat{Q}}^{\ell }{\hat{U}}_{S}^{-1}\left(t^{\prime },t_{\ell }\right)=\sum_{m_{\ell }}Q_{m_{\ell }}^{\ell }{\hat{\pi }}_{m_{\ell }}\left(t_{\ell },t^{\prime }\right), \end{array}$$
where ${\hat{\pi }}_{m_{\ell }}\left(t_{\ell },t^{\prime }\right)={\hat{U}}_{S}\left(t^{\prime },t_{\ell }\right){\hat{\pi }}_{m_{\ell }}{\hat{U}}_{S}^{-1}\left(t^{\prime },t_{\ell }\right)$ is the projector ${\hat{\pi }}_{m_{\ell }}$ evolved *backwards* in time from $t^{\prime }$ to $t_{\ell }$. To the second kind belong the quantities
(27)$$\begin{array}{c} {\hat{Q}}^{+}\left(t^{\prime }\right)=\sum_{m_{+}}Q_{m_{+}}^{+}{\hat{\pi }}_{m_{+}}^{+}\equiv {\hat{U}}_{S}^{-1}\left(t_{\ell +1},t^{\prime }\right){\hat{Q}}^{\ell +1}{\hat{U}}_{S}\left(t_{\ell +1},t^{\prime }\right)\equiv \sum_{m_{\ell +1}}Q_{m_{\ell +1}}^{\ell +1}{\hat{\pi }}_{m_{\ell +1}}\left(t_{\ell +1},t^{\prime }\right) \end{array}$$
where ${\hat{\pi }}_{m_{\ell +1}}\left(t_{\ell +1},t^{\prime }\right)={\hat{U}}_{S}^{-1}\left(t_{\ell +1},t^{\prime }\right){\hat{\pi }}_{m_{\ell +1}}{\hat{U}}_{S}\left(t_{\ell +1},t^{\prime }\right)$ is the projector ${\hat{\pi }}_{m_{\ell +1}}$ evolved *forwards* in time from $t^{\prime }$ to $t_{\ell +1}$. Indeed, since ${\hat{\pi }}_{m_{\ell }}{\hat{\pi }}_{m_{\ell }^{\prime }}={\hat{\pi }}_{m_{\ell }}\delta_{m_{\ell }m_{\ell }^{\prime }}$ and ${\hat{\pi }}_{m_{\ell +1}}{\hat{\pi }}_{m_{\ell +1}^{\prime }}={\hat{\pi }}_{m_{\ell +1}}\delta_{m_{\ell +1}m_{\ell +1}^{\prime }}$, we have
(28)$$\begin{array}{c} {\hat{\pi }}_{m_{\ell +1}}{\hat{U}}_{S}\left(t_{\ell +1},t^{\prime }\right){\hat{\pi }}_{m_{\ell }}\left(t_{\ell },t^{\prime }\right){\hat{U}}_{S}\left(t^{\prime },t_{\ell }\right){\hat{\pi }}_{m_{\ell }}=\;\;\;\\ {\hat{\pi }}_{m_{\ell +1}}{\hat{U}}_{S}\left(t_{\ell +1},t^{\prime }\right){\hat{\pi }}_{m_{\ell +1}}\left(t_{\ell +1},t^{\prime }\right){\hat{U}}_{S}\left(t^{\prime },t_{\ell }\right){\hat{\pi }}_{m_{\ell }}={\hat{\pi }}_{m_{\ell +1}}{\hat{U}}_{S}\left(t_{\ell +1},t_{\ell }\right){\hat{\pi }}_{m_{\ell }}, \end{array}$$
and [cf. Equation (Equation 6)]
(29)$$\begin{array}{c} P(Q_{m_{L}}^{L}\ldots \leftarrow Q_{m_{\ell +1}}^{\ell +1}\leftarrow Q_{m_{\ell }}^{\ell }\ldots \leftarrow q_{n_{0}}^{0})=\;\;\;\\ P(Q_{m_{L}}^{L}\ldots \leftarrow Q_{m_{\ell +1}}^{\ell +1}\leftarrow Q_{m_{\ell }}^{-}\left(t^{\prime }\right)\leftarrow Q_{m_{\ell }}^{\ell }\ldots \leftarrow q_{n_{0}}^{0})=\\ P(Q_{m_{L}}^{L}\ldots \leftarrow Q_{m_{\ell +1}}^{\ell +1}\leftarrow Q_{m_{\ell +1}}^{+}\left(t^{\prime }\right)\leftarrow Q_{m_{\ell }}^{\ell }\ldots \leftarrow q_{n_{0}}^{0}). \end{array}$$
There is, of course, a simple explanation. The states ${\hat{U}}_{S}\left(t^{\prime },t_{\ell }\right)|q_{n_{\ell }}^{\ell }\rangle$ form an orthogonal basis for measuring ${\hat{Q}}^{-}\left(t^{\prime }\right)$, and the system in $|q_{n_{\ell }}^{\ell }\rangle$ at $t_{\ell }$ can only go to $|{\hat{U}}_{S}\left(t^{\prime },t_{\ell }\right)|q_{n_{\ell }}^{\ell }\rangle$ at $t^{\prime }$, as all other matrix elements of ${\hat{U}}_{S}\left(t^{\prime },t_{\ell }\right)$ vanish. Similarly, the system in ${\hat{U}}_{S}^{-1}\left(t_{\ell +1},t^{\prime }\right)|q_{n_{\ell +1}}^{\ell }\rangle$ at $t^{\prime }$ can only go to $|q_{n_{\ell +1}}^{\ell }\rangle$ at $t_{\ell +1}$. The presence of the operators ${\hat{U}}_{S}^{-1}\left(t^{\prime },t_{\ell }\right)$ and ${\hat{U}}_{S}\left(t_{\ell +1},t^{\prime }\right)$ in Equations (Equation 26) and (Equation 27) ensures that Equation (Equation 25) holds, and Equation (Equation 29) follow.

The problem is as follows. By using the CR, we appear to be able to say that at $t=t^{\prime }$ a quantity ${\hat{Q}}^{-}\left(t^{\prime }\right)$ has a definite value $Q_{m_{\ell }}^{-}$ if the value of ${\hat{Q}}^{\ell }$ at $t=t_{\ell }$ was $Q_{m_{\ell }}^{\ell }$. Similarly, it would appear that ${\hat{Q}}^{+}\left(t^{\prime }\right)$ also has a definite value $Q_{m_{\ell +1}}^{+}$ if the value of ${\hat{Q}}^{\ell +1}$ at $t=t_{\ell +1}$ is $Q_{m_{\ell +1}}^{\ell +1}$ Since in general ${\hat{Q}}^{-}\left(t^{\prime }\right)$ and ${\hat{Q}}^{+}\left(t^{\prime }\right)$ do not commute, $[{\hat{Q}}^{-}\left(t^{\prime }\right),{\hat{Q}}^{+}\left(t^{\prime }\right)]\neq 0$, and quantum mechanics forbids ascribing simultaneous values to non-commuting quantities, we seem to have a contradiction.

Fortunately, the contradiction is easily resolved. At the end of the experiment, one needs to have all the relevant records and to produce these records an additional probe must be coupled at $t=t^{\prime }$. Measuring ${\hat{Q}}^{-}\left(t^{\prime }\right)$, or ${\hat{Q}}^{+}\left(t^{\prime }\right)$ requires different probes, which affect the system differently, and produce different statistical ensembles. The values $Q_{m_{\ell }}^{-}$ and ${\hat{Q}}_{m_{\ell +1}}^{+}$ do not pre-exist in their respective measurements [9,21], and appear as a result of a probe acting on a system. The caveat is the same as in Bohr’s answer [25] to the authors of [24]. There are no practical means of ascertaining these conflicting values *simultaneously*. Next, we give a simple example.

## 6. A Two-Level Example

Consider a two-level system, (a qubit), N=2, prepared by the first measurement of an operator
(30)$$\begin{array}{c} {\hat{Q}}^{0}\equiv \hat{B}=\sum_{i=1}^{2}\left|b_{i}\rangle B_{i}\langle b_{i}\right| \end{array}$$
in a state $|b_{1}\rangle$ at $t=t_{0}$. The second measurement (we have L=1) yields one of the eigenvalues of an operator
(31)$$\begin{array}{c} {\hat{Q}}^{1}\equiv \hat{C}=\sum_{i=1}^{2}\left|c_{i}\rangle C_{i}\langle c_{i}\right|,\;\left[\hat{B},\hat{C}\right]\neq 0, \end{array}$$
With only two dimensions involved, all eigenvalues are non-degenerate. If for simplicity we put ${\hat{H}}_{S}=0$, ${\hat{U}}_{S}\left(t_{1},t_{0}\right)=1$ (see Figure 5a), one can easily verify that at any $t_{0}<t^{\prime }<t_{1}$ the value of $\hat{B}$ is $B_{1}$ (see Figure 5b), or that $\hat{C}$ has the same value it will have at $t=t_{1}$ (see Figure 5c). Moreover, its is easy to ascertain that if the first and the last outcomes are $B_{1}$ and $C_{i}$, the values of $\hat{B}$ at $t=t^{\prime }$ and of $\hat{C}$ at $t=t^{\prime \prime }$ are $B_{1}$ and $C_{i}$, as long as $t_{0}<t^{\prime }<t^{\prime \prime }<t_{1}$ (see Figure 6a). Indeed, according to (Equation 3) we have
(32)$$\begin{array}{c} P\left(C_{1}\leftarrow C_{1}\leftarrow B_{1}\leftarrow B_{1}\right)=|\langle c_{1}|c_{1}\rangle \langle c_{1}|b_{1}\rangle \langle b_{1}|b_{1}{\rangle |}^{2}=|\langle c_{1}{\left|b_{1}\rangle \right|}^{2}=P\left(C_{1}\leftarrow B_{1}\right). \end{array}$$
The former is no longer true if $\hat{C}$ is measured before $\hat{B}$. A final value $C_{1}$ can now be reached via four real paths $\left\{C_{1}\leftarrow B_{i}\leftarrow C_{j}\leftarrow B_{1}\right\}$ shown in Figure 6b, and the probabilities no longer agree with those Equation (Equation 32),
(33)$$\begin{array}{c} \tilde{P}\left(C_{1}\leftarrow B_{1}\right)\equiv \sum_{i,j=1}^{2}P\left(C_{1}\leftarrow B_{i}\leftarrow C_{j}\leftarrow B_{1}\right)\neq P\left(C_{1}\leftarrow B_{1}\right). \end{array}$$
The transition between the two regimes occurs at $t^{\prime }=t^{\prime \prime }$, when an attempt is made to measure two non-commuting quantities at the same time. The rules of Section 2 imply that such measurements are not possible in principle, since $\hat{B}$ and $\hat{C}$ do not have a joint set of eigenstates which could inserted into Equation (Equation 2). However, ultimately one is interested in the records available at the end of the experiment. Next, we will look at the readings the probes would produce, should they be set up to measure non-commuting $\hat{B}$ and $\hat{C}$ simultaneously.

## 7. Joint Measurement of Non-Commuting Variables

We want to consider two measurements made on the system in Figure 6 which can overlap in time at least partially. No longer instantaneous, both measurement will last τ seconds, start at $t^{\prime }$ and $t^{\prime \prime }$, $t^{\prime },t^{\prime \prime }>t_{0}$, respectively, and finish before $t=t_{1}$, $t^{\prime }+\tau ,t^{\prime \prime }+\tau <t_{1}$. The degree to which the measurements overlap will be controlled by a parameter β,
(34)$$\begin{array}{c} \beta =(t^{\prime \prime }-t^{\prime })/\tau , \end{array}$$
so that for β=1 the measurement of $\hat{B}$ precedes that of $\hat{C}$, β=0 corresponds to simultaneous measurements of both $\hat{B}$ and $\hat{C}$, and for β=−1
$\hat{C}$ is measured first. Next we consider the two kind of probes introduced in Section 4 separately.

### 7.1. C-NOT Gates as a Meters

For our two-level example of Section 6 we can further simplify the probe described in Section 4. Since we only need to distinguish between two of the system’s conditions, a two-level probe, whose state either changes or remains the same, is all that is required. We will need two such probes, $D^{\prime }$ and $D^{\prime \prime }$, two sets of states
(35)$$\begin{array}{c} |D^{\ell }\left(1\right)\rangle =|1^{\ell }\rangle ,\;\left|D^{\ell }\left(2\right)\rangle =\right|2^{\ell }\rangle ,\;\ell ={,}^{\prime }{,}^{\prime \prime } \end{array}$$
four projectors
(36)$$\begin{array}{c} {\hat{\pi }}_{1}^{\prime }=|b_{1}\rangle \langle b_{1}|,\;{\hat{\pi }}_{2}^{{}^{\prime }}=|b_{2}\rangle \langle b_{2}|,\;{\hat{\pi }}_{1}^{{}^{\prime \prime }}=|c_{1}\rangle \langle c_{1}|,\;{\hat{\pi }}_{2}^{{}^{\prime \prime }}=\left|c_{2}\rangle \langle c_{2}\right|. \end{array}$$
and two couplings
(37)$$\begin{array}{c} {\hat{H}}_{int}^{\prime }=-\left(\pi /2\right)\tau^{-1}{\hat{\pi }}_{2}^{{}^{\prime }}{\hat{\sigma }}_{x}^{{}^{\prime }},\;{\hat{H}}_{int}^{{}^{\prime \prime }}=-\left(\pi /2\right)\tau^{-1}{\hat{\pi }}_{2}^{{}^{\prime \prime }}{\hat{\sigma }}_{x}^{{}^{\prime \prime }}, \end{array}$$
where ${\hat{\sigma }}_{x}^{\ell }\left|D^{\ell }\left(1\right)\rangle =\right|D^{\ell }\left(2\right)\rangle$. In what follows we will put τ=1. The probes are prepared in the respective states $|D^{\prime }\left(1\right)\rangle$ and $|D^{\prime \prime }\left(1\right)\rangle$, and after finding the system in $|c_{2}\rangle$ at $t_{2}$ their state is given by
(38)$$\begin{array}{c} |\Phi_{Probes}\left(t_{2}\right)\rangle \equiv \;\;\;\;\;\;\;\;\;\;\;\;\;\;\;\;\;\;\;\;\;\;\;\;\\ \langle c_{1}|\exp[-i|\beta |{\hat{H}}_{int}^{{}^{\prime \prime }}]⊗\exp[-i(1-|\beta |)\left({\hat{H}}_{int}^{{}^{\prime \prime }}+{\hat{H}}_{int}^{{}^{\prime }}\right)]⊗\exp[-i|\beta |{\hat{H}}_{int}^{{}^{\prime }}]|b_{1}\rangle \left|D^{\prime }\left(1\right)\rangle \right|D^{{}^{\prime \prime }}\left(1\right)\rangle .\;\; \end{array}$$
if $t^{\prime \prime }>t^{\prime }$, while for $t^{\prime \prime }<t^{\prime }$ the order of operators in (Equation 38) is reversed. It is easy to check that ($\ell ={,}^{\prime }{,}^{\prime \prime }$)
(39)$$\begin{array}{c} \exp[-i|\beta |{\hat{H}}_{int}^{\ell }]=\left\{{\hat{\pi }}_{1}^{\ell }+{\hat{\pi }}_{2}^{\ell }\left[\cos(\pi |\beta |/2)+i\sin(\pi |\beta |/2\right){\hat{\sigma }}_{x}^{\ell }]\right\} \end{array}$$
so that for |β|=1 the r.h.s. of Equation (Equation 39) reduces to ${\hat{\pi }}_{1}^{\ell }+i{\hat{\pi }}_{2}^{\ell }\sigma_{x}^{\ell }$. The action of the coupling is, therefore, that of a quantum (C)ontrolled-NOT gate [26], which flips the probe’s (target) state if the system’s (control) state is $|b_{2}\rangle$ or $|c_{2}\rangle$, and leaves the probes’s condition unchanged if it is $|b_{1}\rangle$ or $|c_{1}\rangle$.

There are four possible outcomes, $\left(1^{\prime },1^{\prime \prime }\right)$, $\left(1^{\prime },2^{\prime \prime }\right)$, $\left(2^{\prime },1^{\prime \prime }\right)$, and $\left(2^{\prime },2^{\prime \prime }\right)$, and four corresponding probabilities (*i*, *j* = 1, 2),
(40)$$\begin{array}{c} P\left(i^{\prime },j^{\prime \prime }\right)=|\langle D^{\prime }\left(i\right)|\langle D^{\prime \prime }\left(j\right)|\Phi_{Probes}\left(t_{2}\right){\rangle |}^{2}/\sum_{k,l=1}^{2}|\langle D^{\prime }\left(k\right)|\langle D^{\prime \prime }\left(l\right){\left|\Phi_{Probes}\left(t_{2}\right)\rangle \right|}^{2},\;. \end{array}$$
The matrix elements are easily evaluated (for details see Appendix B), and the results are shown in Figure 7.

If β=1, there is no overlap, and $\hat{B}$ is measured before $\hat{C}$. Dividing τ into *K* sub-intervals and sending K→∞ we have an identity (cf. Equation (Equation 39))
(41)$$\begin{array}{c} \exp\left[-i{\hat{H}}_{int}^{{}^{\prime }}\right]|b_{1}\rangle ={\left[\exp(-i{\hat{H}}_{int}^{{}^{\prime }}/K])\right]}^{K}|b_{1}\rangle ={\left({\hat{\pi }}_{1}^{\prime }\right)}^{K}\left|b_{1}\rangle =\right|b_{1}\rangle , \end{array}$$
and the state of the first probe remains unchanged. For β=0 both probes act simultaneously. Now the use of the Trotter’s formula yields
(42)$$\begin{array}{c} \exp\left[-i\left({\hat{H}}_{int}^{{}^{\prime }}+{\hat{H}}_{int}^{{}^{\prime \prime }}\right)\right]\left|b_{1}\rangle =\lim_{K\to \infty }{\left(1+i\pi {\hat{\pi }}_{2}^{{}^{\prime }}{\hat{\sigma }}_{x}^{{}^{\prime }}/2K+i\pi {\hat{\pi }}_{2}^{{}^{\prime \prime }}{\hat{\sigma }}_{x}^{{}^{\prime \prime }}/2K\right)}^{K}\right|b_{1}\rangle . \end{array}$$
Equation (Equation 42) contains scenarios where both probes change their states, and since ${\hat{\pi }}^{{}^{\prime }}⊗{\hat{\pi }}^{{}^{\prime \prime }}\neq 0$, the evolution of one of them must affect what happens to the other. Now all four probabilities in Equation (Equation 40) have non-zero values, although it is still more likely that both probes will remain in their initial states (cf. Figure 7). Finally, for β=−1, $\hat{C}$ is measured before $\hat{B}$ as if both measurements were instantaneous, and all four paths in Figure 6b are equally probable. A different result is obtained if the two measurements are of von Neumann type, as we will discuss next.

### 7.2. Von Neumann Meters

Consider the same problem but with the two-level probes replaced by two von Neumann pointers with positions $f^{\prime }$ and $f^{\prime \prime }$, respectively. As before, the interaction with each pointer lasts τ seconds, so the two Hamiltonians are
(43)$$\begin{array}{c} {\hat{H}}_{int}^{\prime }=-i\partial_{f^{\prime }}\hat{B}/\tau ,\;{\hat{H}}_{int}^{\prime \prime }=-i\partial_{f^{\prime \prime }}\hat{C}/\tau . \end{array}$$
If the pointers are prepared in identical Gaussian states (Equation 22) $\langle f^{\ell }|G^{\ell }\rangle =G\left(f^{\ell }\right),\;\ell ={,}^{\prime }{,}^{\prime \prime }$ the probability distribution of their final positions is given by
(44)$$\begin{array}{c} P\left(f^{\prime },f^{\prime \prime }\right)={\left|\int dy^{\prime }dy^{\prime \prime }G\left(f^{\prime }-y^{\prime }\right)G\left(f^{\prime \prime }-y^{\prime \prime }\right)\Phi \left(y^{\prime },y^{\prime \prime }\right)\right|}^{2}\equiv {\left|\Psi \left(f^{\prime },f^{\prime \prime }\right)\right|}^{2} \end{array}$$
where (we measure *f* in units of $g_{0}$ and put τ=1)
(45)$$\begin{array}{c} \Phi \left(f^{\prime },f^{\prime \prime }\right)\equiv \langle c_{1}|\langle f^{\prime }|\langle f^{\prime \prime }|\exp\left[-|\beta \partial_{f^{\prime \prime }}\hat{C}\right]⊗\exp[-(1-2|\beta |)\left(\partial_{f^{\prime }}\hat{B}+\partial_{f^{\prime \prime }}\hat{C}\right)]\\ ⊗\exp[-|\beta |\partial_{f^{\prime \prime }}\hat{B}]|0^{\prime }\rangle \left|0^{\prime \prime }\rangle \right|b_{1}\rangle \end{array}$$
where $\langle f^{\prime }|0^{\prime }\rangle =\delta \left(f^{\prime }\right)$, $\langle f^{\prime \prime }|0^{\prime \prime }\rangle =\delta \left(f^{\prime \prime }\right)$, and $t^{\prime }\leq t^{\prime \prime }$. (For $t^{\prime }>t^{\prime \prime }$ we interchange $\hat{B}$ with $\hat{C}$.)
Consider first the case β=0 where the measurements coincide. The amplitude $\Phi (f^{\prime },f^{\prime \prime })$ has several general properties. Firstly, it cannot be a smooth finite function of $y^{\prime }$ and $y^{\prime \prime }$, or the integral in (Equation 44) would vanish in the limit of narrow Gaussians, Δf→0, due to the normalisation of $|G^{\prime }\rangle$ and $|G^{\prime \prime }\rangle$ (cf. Equation (Equation 22)). It must, therefore, have δ-singularities [27,28]. Secondly, using the Trotter’s formula [29], we have
(46)$$\begin{array}{c} \exp\left[-\left(\partial_{f^{\prime }}\hat{B}+\partial_{f^{\prime \prime }}\hat{C}\right)\right]=\lim_{K\to \infty }{\left[\exp\left(-\partial_{f^{\prime }}\hat{B}/K\right)⊗\exp\left(-\partial_{f^{\prime \prime }}\hat{C}/K\right)\right]}^{K} \end{array}$$
It is readily seen that each time the product in the r.h.s. of Equation (Equation 46) is applied, the pointers are displaced by $B_{1}/K$ or $B_{2}\tau /K$ and $C_{1}/K$ or $C_{2}/K$, respectively. If $B_{1,2},C_{1,2}=\pm 1$, one has a quantum random walk, where the pointers are shifted by an equal amount 1/K ether to the right or to the left. Since the largest possible displacement is 1, $\Phi (f^{\prime },f^{\prime \prime })$ must vanish outside a square $-1\leq f^{\prime },f^{\prime \prime }\leq 1$. One also notes that for $f^{\prime }=\pm 1$, the maximum of $\Phi (\pm 1,f^{\prime \prime })$ is reached for $f^{\prime \prime }=0$, since there are (let *K* be an even number) K!/(K/2)!(K/2)! walks, each contributing to $\Phi (f^{\prime },f^{\prime \prime })$ the same amount $1/2^{2K+1}$. Similarly, a maximum of $\Phi (f^{\prime },\pm 1)$ is reached for $f^{\prime }=0$.

A detailed combinatorial analysis is complicated, but the location of the singularities can be determined as was done in [30]. As explained in the Appendix B, the amplitude $\Phi (y^{\prime },y^{\prime \prime })$ in Equation (Equation 45) can also be written as ($f\equiv \sqrt{f^{\prime 2}+f^{\prime \prime 2}}$, $\cos\left(\theta \right)=f^{\prime }/f$)
(47)$$\begin{array}{c} \Psi \left(f^{\prime },f^{\prime \prime }\right)={\left(2\pi \right)}^{-1/2}\Delta f\langle c_{1}|b_{1}\rangle \int_{0}^{\infty }d\lambda \lambda \exp\left(-\lambda^{2}\Delta f^{2}/4\right)\times \\ \left[\cos\left(\lambda \right)J_{0}\left(\lambda f\right)-\sqrt{2}i\sin\left(\lambda \right)J_{1}\left(\lambda f\right)\sin\left(\theta -\pi /4\right)\right] \end{array}$$
where $J_{k}$ is the Bessel function of the first kind of order *k*. From Equation (Equation 44) one notes that in the limit Δf→0, $P(f^{\prime },f^{\prime \prime })$ in Equation (Equation 44) will become singular, and that its singularities will coincide with those of $\Phi (f^{\prime },f^{\prime \prime })$ in Equation (Equation 45). Since [31]
(48)$$\begin{array}{c} J_{k}\left(\lambda f\to \infty \right)\to {\left(2/\pi \lambda f\right)}^{1/2}\cos\left(\lambda f-k\pi /2-\pi /2\right), \end{array}$$
the integral in Equation (Equation 47) will diverge at large λ’s provided the oscillations of cos(λ) and sin(λ) are cancelled by those of $J_{0}\left(\lambda f\right)$ and $J_{1}\left(\lambda f\right)$, i.e., for $\sqrt{f^{\prime 2}+f^{\prime \prime 2}}=1$. As a result, we find the pointer readings of two simultaneous accurate measurements of $\hat{B}$ and $\hat{C}$ (with $B_{1,2}=C_{1,2}=\pm 1$) distributed along the perimeter of a unit circle as shown in Figure 8. Figure 9 shows the distribution of the pointer’s readings for different degrees of overlap between the two measurements.

There is, therefore, an important difference between employing discrete probes and von Neumann pointers. In the previous example shown in Figure 7 one could (although should not) assume, e.g., that for β=0
$P(1^{\prime },2^{\prime \prime })$ yields the probability for $\hat{B}$ and $\hat{C}$ to have the values 1 and −1, if measured simultaneously. Figure 8 and Figure 9 show this conclusion to be inconsistent. As β decreases from 1 to −1, the pointer’s readings are not restricted to $\pm 1^{\prime }$, $\pm 1^{\prime \prime }$. Rather, for β=0 they are along the perimeters of a unit circle (cf. Figure 8 and Figure 9), which, if taken at face value, implies that the probability in question is zero. We already noted that the theory in Section 2 is unable to prescribe probabilities of simultaneous values of non-commuting operators. In practice, this means that the probes, capable of performing the task in a consistent manner, simply cannot be constructed.

## 8. The Past of a Quantum System: Weakly Perturbing Probes and the Uncertainty Principle

It remains to see what information can be obtained from the measurements, designed to perturb the measured system as little as possible. If such a measurement were an attempt to distinguish between interfering scenarios, without destroying interference between them, it would contradict the Uncertainty Principle cited in Section 2. As before, we treat the two types of probes separately.

### 8.1. Weak Discrete Gates

We start by reducing the coupling strength, so the interaction Hamiltonians (Equation 15) become
(49)$$\begin{array}{c} {\hat{H}}_{int}^{\ell }=-\gamma \sum_{m_{\ell }=1}^{M_{\ell }}{\hat{\pi }}_{m_{\ell }}^{\ell }{\hat{\sigma }}_{x}\left(m_{\ell }\right)\delta \left(t-t_{\ell }\right),\;\gamma <\pi /2. \end{array}$$
In [11], Feynman described a double-slit experiment where photons, scattered by the the passing electron, allowed one to know through which of the two slits the electron has travelled. With every electron duly detected, their distribution on the screen, $P_{\mathrm{Slit}-\mathrm{unknown}}\left(x\right)$, does not exhibit an interference pattern. With no photons present, the pattern is present in the distribution $P_{\mathrm{Slit}-\mathrm{known}}\left(x\right)$. If the intensity of light (i.e., the number of photons) is decreased, some of the electrons pass undetected. The total distribution on the screen is, therefore, “a mixture of the two curves” [11]. $P\left(x\right)=aP_{\mathrm{Slit}-\mathrm{unknown}}\left(x\right)+bP_{\mathrm{Slit}-\mathrm{known}}\left(x\right)$, where *a* and *b* are some constants.

Something very similar happens if an extra discrete probe $D^{\prime }$ is added to measure ${\hat{Q}}^{\prime }=\sum_{m^{\prime }=1}^{M^{\prime }}Q_{m^{\prime }}^{\prime }{\hat{\pi }}_{m^{\prime }}^{\prime }$ at $t=t^{\prime }$, $t_{\ell }<t^{\prime }<t_{\ell +1}$. As before [cf Equation (Equation 16)], we find
(50)$$\begin{array}{c} {\hat{U}}_{int}^{\prime }\left(t^{\prime }\right)=\exp\left[i\gamma \sum_{m^{\prime }=1}^{M^{\prime }}{\hat{\pi }}_{m^{\prime }}^{\prime }{\hat{\sigma }}_{x}^{\prime }\left(m^{\prime }\right)\right]=\cos\left(\gamma \right)+i\sin\left(\gamma \right)\sum_{m^{\prime }l=1}^{M^{\prime }}{\hat{\pi }}_{m^{\prime }}^{\prime }{\hat{\sigma }}_{x}^{\prime }\left(m^{\prime }\right), \end{array}$$
where the cosine term accounts for the possibility that the systems passes the check undetected. Replacing P(x) in Feynman’s example with the probability $P(q_{n_{L}}^{L})$ of detecting the system in a final state $|q_{n_{L}}^{L}\rangle$ at $t_{L}$. With a values of ${\hat{Q}}^{\prime }$ detected in every run, γ=π/2, one has a distribution
(51)$$\begin{array}{c} P_{Q^{\prime }-\mathrm{known}}\left(q_{n_{L}}^{L}\right)=\sum_{m^{\prime }=1}^{M^{\prime }}{\left|A_{S}\left(q_{n_{L}}^{L}\ldots \leftarrow {\hat{\pi }}_{m_{\ell +1}}^{\ell +1}\leftarrow {\hat{\pi }}_{m^{\prime }}^{\prime }\leftarrow {\hat{\pi }}_{m_{\ell }}^{\ell }\ldots \leftarrow q_{n_{0}}^{0}\right)\right|}^{2} \end{array}$$
where $A_{S}$ is the amplitude in Equation (Equation 5). If the probe is uncoupled, γ=0, the distribution is
(52)$$\begin{array}{c} P_{Q^{\prime }-\mathrm{unknown}}\left(q_{n_{L}}^{L}\right)={\left|\sum_{m^{\prime }=1}^{M^{\prime }}A_{S}\left(q_{n_{L}}^{L}\ldots \leftarrow {\hat{\pi }}_{m_{\ell +1}}^{\ell +1}\leftarrow {\hat{\pi }}_{m^{\prime }}^{\prime }\leftarrow {\hat{\pi }}_{m_{\ell }}^{\ell }\ldots \leftarrow q_{n_{0}}^{0}\right)\right|}^{2} \end{array}$$
With 0<γ<π/2 the outcomes fall into two groups, those where the probe $D^{\prime }$ remains in its initial state, and those where the state of one of its sub-systems has been flipped. The two alternatives are exclusive [32], and the total distribution is indeed a mixture of the two curves. As γ→0 one has
(53)$$\begin{array}{c} P\left(q_{n_{L}}^{L}\right)=\left(1-\gamma^{2}\right)P_{Q^{\prime }-\mathrm{unknown}}\left(q_{n_{L}}^{L}\right)+\gamma^{2}P_{Q^{\prime }-\mathrm{known}}\left(q_{n_{L}}^{L}\right)+o\left(\gamma^{2}\right). \end{array}$$

In other words, in the vast majority of cases, the system remains undetected, and the interference is preserved (cf. Equation (Equation 52)). In the few remaining cases, it is detected and the interference is destroyed. The Uncertainty Principle [1] is obeyed to the letter: probabilities are added where records allow one to distinguish between the scenarios; otherwise one sums the amplitudes. One has, however, to admit that nothing really new has been learned from this example, as both possibilities simply illustrate the rules of Section 2.

The above analysis is easily extended to include more extra measurements, whether impulsive or not. Since to the first order in the coupling constant γ weak probes act independently of each other, the r.h.s. of Equation (Equation 53) would contain additional terms $P_{Q^{\prime \prime }-\mathrm{known}}\left(q_{n_{L}}^{L}\right)$, $P_{Q^{\prime \prime \prime }-\mathrm{known}}\left(q_{n_{L}}^{L}\right)$, etc.

### 8.2. Weak von Neumann Pointers

Next we add to L+1 accurate impulsive von Neumann pointers an extra “weak” pointer, designed to measure ${\hat{Q}}^{\prime }=\sum_{m^{\prime }=1}^{M^{\prime }}Q_{m^{\prime }}^{\prime }{\hat{\pi }}_{m^{\prime }}^{\prime }$ at $t=t^{\prime }$ between $t_{\ell }$ and $t_{\ell +1}$. The new coupling, given by
(54)$$\begin{array}{c} {\hat{H}}_{int}^{\prime }=-i\gamma \partial_{f^{\prime }}{\hat{Q}}^{\prime }, \end{array}$$
will perturb the system only slightly in the limit γ→0. To see what happens in this limit, one can replace $f^{\prime }$ by $\gamma f^{\prime }$, ${\hat{H}}_{int}^{\ell }$ in Equation (Equation 54) by $-i\partial_{f^{\prime }}{\hat{Q}}^{\ell }$, and the pointer’s initial state $G^{\prime }\left(f^{\prime }\right)$ by $\gamma^{1/2}G^{\prime }\left(\gamma f^{\prime }\right)$. Now as γ→0 the pointer’s initial states become very broad, while the coupling remains unchanged. For a Gaussian pointers (Equation 22), considered here, this means replacing $\Delta f^{\prime }$ with $\Delta f^{\prime }/\gamma$, i.e., making the measurement highly inaccurate. This makes sense. The purpose of a pointer is to destroy interference between the system’s virtual paths, (cf. Equation (Equation 24)), which it is clearly unable to do if the coupling vanishes. Accordingly, with the pointer’s initial position highly uncertain, its final reading is also spread almost evenly between −∞ and *∞*. Measured in this manner, the value of ${\hat{Q}}^{\prime }$ remains indeterminate, as required by the Uncertainty Principle.

This could be the end of our discussion, except for one thing. It is still possible to use the broad distribution of a pointer’s readings in order to evaluate averages, which could, in principle, remain finite in the limit $\Delta f^{\prime }\to \infty$. Maybe this can tell us something new about the system’s condition at $t=t^{\prime }$. Note, however, that whatever information is extracted in this manner should not contradict the Uncertainty Principle, or the whole quantum theory would be in trouble [1].

From Equation (Equation 22) it is already clear that any average of this type will be expressed in terms of the amplitudes $A(q_{n_{L}}^{L}\ldots \leftarrow {\hat{\pi }}_{m_{\ell }}^{\ell }\ldots \leftarrow q_{n_{0}}^{0})$. The simplest average is the pointer’s mean position. If the outcomes of the accurate measurements are $Q_{n_{L}}^{L}\ldots Q_{m_{\ell }}^{\ell }\ldots Q_{n_{0}}^{0}$, for the mean reading of the weakly coupled pointer we obtain (see Appendix C)
(55)〈f′(QnLL…Qmℓℓ…Qn00)〉≈Re∑m′=1M′Qm′′AS(qnLL….←π^mℓ+1ℓ+1←π^m′′←π^mℓℓ…←qn00)∑m′=1M′AS(qnLL….←π^mℓ+1ℓ+1←π^m′′←π^mℓℓ…←qn00).
If the measured operator is one of the projectors, say, ${\hat{Q}}^{\ell }={\hat{\pi }}_{m^{\prime }}^{\ell }$, this reduces to
(56)〈f′(QnLL…Qmℓℓ…Qn00)〉≈ReAS(qnLL….←π^mℓ+1ℓ+1←π^m′←π^mℓℓ…←qn00)∑m′=1M′AS(qnLL….←π^mℓ+1ℓ+1←π^m′′←π^mℓℓ…←qn00)
We note that, as in the previous example, different pointers do not affect each other to the leading order in the small parameter γ.

The quantities in the l.h.s. of Equations (Equation 55) and (Equation 56) are the standard averages of the probes’ variables. In the l.h.s. of these equations one finds probability amplitudes for the system’s entire paths, $\left\{q_{n_{L}}^{L}\ldots .\leftarrow {\hat{\pi }}_{m_{\ell +1}}^{\ell +1}\leftarrow {\hat{\pi }}_{m^{\prime }}^{\prime }\leftarrow {\hat{\pi }}_{m_{\ell }}^{\ell }\ldots \leftarrow q_{n_{0}}^{0}\right\}$. The values of these amplitudes can be deduced from the probes’s probabilities [33]. The problem is, these values offer no insight into the condition of the system at $t=t_{\ell }$. In the double-slit case, to conclude that a particle *“...goes through one hole or the other when you are not looking is to produce an error in prediction”* [11]. In our case, one cannot say that the value of ${\hat{Q}}^{\prime }$ was or was not a particular $Q_{m^{\prime }}^{\prime }$. The Uncertainty Principle prevails again, this time by letting one only gain information not sufficient for determining the condition of the unobserved system at $t=t^{\prime }$. A more detailed discussion of this point can be found, e.g., in [7,8].

## 9. Summary and Conclusions

A very general way to describe quantum mechanics is to say that it is a theory that prescribes probability amplitudes to sequences of events, and then predicts the probability of a sequence by taking an absolute square of the corresponding amplitude [1]. Where several (L+1) consecutive measurements are made on the same system (*S*), a sequence of interest is that of the measured values ${\hat{Q}}_{m_{\ell }}^{\ell }$, endowed with a (system’s) probability amplitude $A_{S}\left(q_{n_{L}}^{L}\ldots \leftarrow {\hat{\pi }}_{m_{\ell }}^{\ell }\ldots \leftarrow q_{n_{0}}^{0}\right)$ in Equation (Equation 5).

This is not, however, the whole story. To test the theory’s predictions, an experimenter (Observer) must keep the records of the measured values, in order to collect the statistics once the experiment is finished. This is more than a mere formality. The system, whose condition changes after each measurement, cannot itself store this information. Hence the need for the probes, material objects, whose conditions must be directly accessible to the Observer at the end. One can think of photons [1,11] devices with or without dials, or Observer’s own memories of the past outcomes [17,18]. The probes must be prepared in suitable initial condition $|\Psi_{Probes}\left(0\right)\rangle$ and be found in one of the orthogonal states $|\Psi_{Probes}\left(n_{L},\ldots m_{\ell }\ldots n_{0}\right)\rangle$ later, with an amplitude $A_{S+Probes}(\Psi_{Probes}\left(n_{L},\ldots m_{\ell }\ldots n_{0}\right)$, $q_{n_{L}}^{L}\leftarrow \Psi_{Probes(0)},q_{n_{0}}^{0})$.

To be consistent, the theory must construct the amplitudes $A_{S}$ and $A_{S+Probes}$ using exactly the same rules, and ensure that $|A_{S+Probes}{|}^{2}={\left|A_{S}\right|}^{2}$. In other words, the experimenter should see a record occurring with a frequency the theory predicts for an isolated (no probes) system, going through its corresponding conditions. This requires the existence of a suitable coupling between the system and the probe. Its choice is not unique, and for a system with a finite-dimensional Hilbert space studied here, two different kinds of probes were discussed in Section 4. The first one is a discrete gate, using the interaction in Equation (Equation 15), while the second is the original von Neumann pointer [20].

Now one can obtain the same probability $P=|A_{S+Probes}{|}^{2}={\left|A_{S}\right|}^{2}$ by considering a unitary evolution of a composite {System+Probes} until the moment the Observer examines their records at the end of the experiment. Or one can consider such an evolution of the system only, but broken every time a probe is coupled to it. For a purist intent on identifying quantum mechanics with unitary evolution (see, e.g., the discussion in [34]), the first way may seem preferable. Yet there is no escaping the final collapse of the composite’s wave function when the stock is taken at the end of the experiment.

It is often simpler to discuss measurements in terms of the measured system’s amplitude, leaving out, but not forgetting, the probes. The rules formulated in Section 2 readily give an answer to any properly asked question, but offer no clues regarding a question which has not been asked operationally. One may try to extend the description of a quantum system’s past by looking for additional quantities whose values could be ascertained without changing the probabilities of the measured outcomes. In general, this is not possible. To find the value of a quantity ${\hat{Q}}^{\prime }$ at a $t^{\prime }$ between to successive measurements, $t_{\ell }<t^{\prime }<t_{\ell +1}$, one needs to connect an extra probe. This would destroy interference between the system’s paths and change other probabilities, leaving the question “what was the value of ${\hat{Q}}^{\prime }$ if was not measured?” without an answer.

There are two seeming exceptions to this rule. If ${\hat{Q}}^{\prime }$ is obtained by evolving backward in time the previously measured ${\hat{Q}}^{\ell }$ (cf. Equation (Equation 26)), call it ${\hat{Q}}^{-}$, its value is certain to equal that of ${\hat{Q}}^{\ell }$, and all other probabilities will remain unchanged, (cf. Equation (Equation 28)). Similarly, the value of a ${\hat{Q}}^{+}$ (cf. Equation (Equation 27)), obtained by the forward evolution of the next measured operator ${\hat{Q}}^{\ell +1}$, will also agree with that of ${\hat{Q}}^{\ell +1}$. It would be tempting to assume that these values represent some observation-free “reality”, were it not for the fact that they cannot be ascertained simultaneously. The two measurements require different probes, each affecting the system in a particular way. The probes frustrate each other if employed simultaneously. It is hardly surprising that different evolution operators ${\hat{U}}_{S+Probes}\left(t_{L},t_{0}\right)$, in Equation (Equation 10) may lead to different outcomes.

One notes also that measuring these two quantities one after another would also leave all other probabilities intact, but only if ${\hat{Q}}^{-}$ is measured first, as shown in Figure 6a. Changing this order results in a completely different statistical ensemble, shown in Figure 6b. The rules of Section 2 say little about what happens if the measurements coincide, except that if ${\hat{Q}}^{-}$ and ${\hat{Q}}^{+}$ do not have common eigenstates, Equation (Equation 5) cannot be applied. One can still analyze the behavior of the two probes at different degrees of overlap to explain why it is impossible to reach consistent conclusions about the simultaneous values of ${\hat{Q}}^{-}$ and ${\hat{Q}}^{+}$. For example, if two discrete gates are used, Figure 7 appears to offer four joint probabilities of having the values B,C=±1. If discrete probes are replaced by von Neumann pointers, the readings shown in Figure 8 suggest that joint values of $\hat{B}$ and $\hat{C}$ should lie on the perimeter of a unit circle, in a clear contradiction with the previous conclusion.

Another way to explore the system’s past beyond what has been established by accurate measurements, is to study its response to a weakly perturbing probe, set up to measure some ${\hat{Q}}^{\prime }$ at an intermediate time $t^{\prime }$. In this limit, the two types of probes produce different effects but, in accordance with the Uncertainty Principle, reveal nothing new that can be added to the rules formulated in Section 2. If the coupling of an additional discrete probe is reduced, trials are divided into two groups. In a (larger) number of cases, the system remains undetected, and interference between its virtual paths passing through different eigenstates of ${\hat{Q}}^{\prime }$ remains intact. In a (smaller) fraction of cases, the value of ${\hat{Q}}^{\prime }$ is accurately determined, and the said interference is destroyed completely. Individual readings of a weak von Neumann pointer extend a range much wider than the region which contains the values of ${\hat{Q}}^{\prime }$, and are in this sense practically random. Its mean position (reading) allows one to learn something about the probability amplitude in Equation (Equation 56), or a combination of such amplitudes as in Equation (Equation 55). The problem is that even after obtaining the values of these amplitudes (and this can be done in practice [33]), one still does not know the value of ${\hat{Q}}^{\prime }$, for the same reason he/she cannot know the slit chosen by an unobserved particle in a double-slit experiment. Quantum probability amplitudes simply do not have this kind of predictive power [1,11].

In summary, quantum mechanics can consistently be seen as a formalism for calculating transition amplitudes by means of evaluating matrix elements of evolution operators. In such a “minimalist” approach (see also [35]), the importance of a wave function, represented by an evolving system’s state, is reduced to that of a convenient computational tool. In the words of Peres [23] (see also [36]) *“... there is no meaning to a quantum state before the preparation of the physical system nor after its final observation (just as there is no ‘time’ before the big bang or after the big crunch).”* This is, however, not a universally accepted view. For example, the authors of [4,5,6] propose a time-symmetric formulation of quantum mechanics, employing not one but two evolving quantum states. We will examine the usefulness of such an approach in future work.

## Figures and Tables

**Figure 1 entropy-24-00877-f001:**
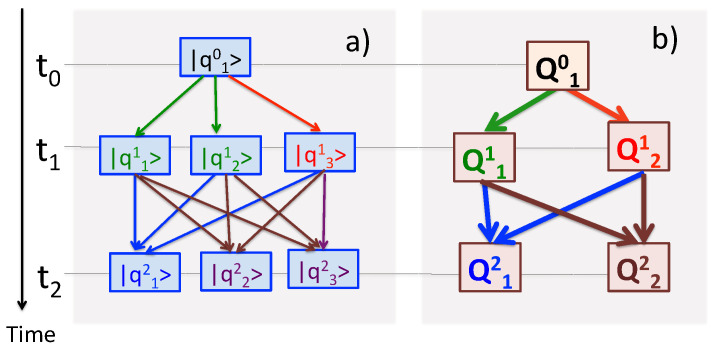
Three measurements, L=3, are made on a three-level system, N=3. The first one, yields an outcome $Q_{1}^{0}$ and prepares the system in a state $|q_{1}^{0}\rangle$. Two other operators have degenerate eigenvalues, ${\hat{Q}}^{1}\left|q_{1,2}^{1}\rangle =Q_{1}^{1}\right|q_{1,2}^{1}\rangle$, ($M_{1}=2$), and ${\hat{Q}}^{2}\left|q_{2,3}^{2}\rangle =Q_{2}^{2}\right|q_{2,3}^{1}\rangle$, ($M_{2}=2$). (**a**) Nine virtual paths in Equation (Equation 3). (**b**) Four real paths (i.e., the observed sequences $Q_{m_{2}}^{2}\leftarrow Q_{m_{1}}^{1}\leftarrow Q_{1}^{0}$), $m_{1},m_{2}=1,2$. Different colours are used to relate the virtual paths to the observed outcomes.

**Figure 2 entropy-24-00877-f002:**
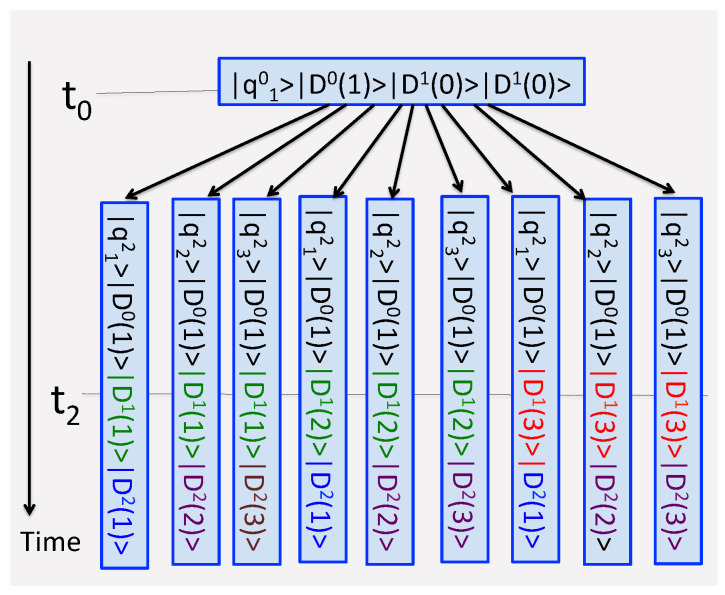
The measurements in Figure 1 seen from a different perspective. Just after $t=t_{2}$, the experimenter needs to compare three records, in order to determine which of the real paths in Figure 1b was actually taken. This information is encoded in the final conditions of the three probes, $D^{\ell }$, ℓ=0,1,2. The composite {system+probes} undergoes an unbroken unitary evolution for $t_{0}\leq t\leq t_{2}$. There are nine virtual paths ending in distinguishable states of the composite. The same colors are used to indicate which of the nine path probabilities should be added to obtain likelihoods of the four real scenarios in Figure 1b.

**Figure 3 entropy-24-00877-f003:**
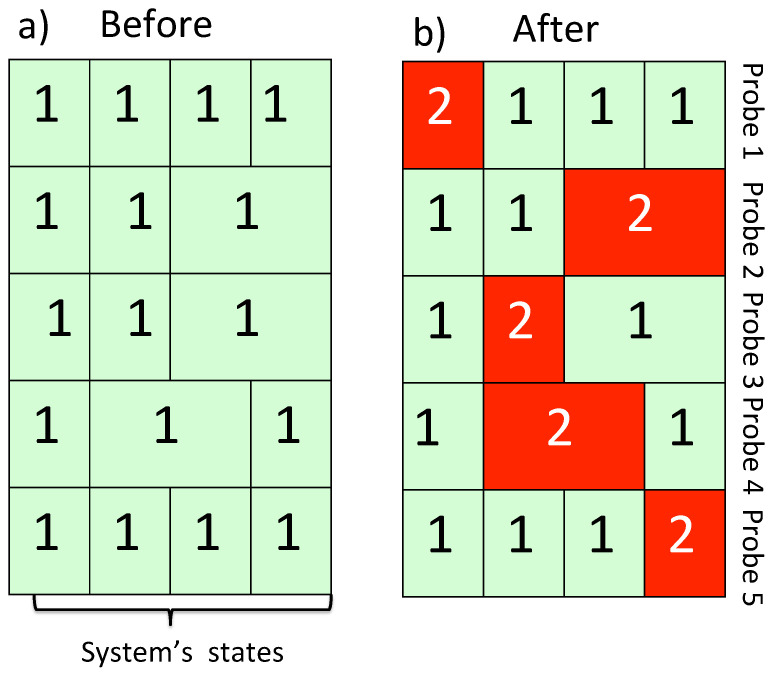
Five consecutive measurements of the quantities ${\hat{Q}}^{0},{\hat{Q}}^{1}\ldots {\hat{Q}}^{4}$ are made on a four-level system (N=4). Some of the eigenvalues are degenerate. Each probe consists of $M_{\ell }\leq 4$ [cf. Equation (Equation 15)] two-level sub-systems. (**a**) Initially sub-systems of all probes are prepared in their lower states |1〉. (**b**) At the end of a trial some these states are found changed, and a record $\left\{Q_{4}^{4}\leftarrow Q_{2}^{3}\leftarrow Q_{2}^{2}\leftarrow Q_{3}^{1}\leftarrow Q_{1}^{0}\right\}$ is produced.

**Figure 4 entropy-24-00877-f004:**
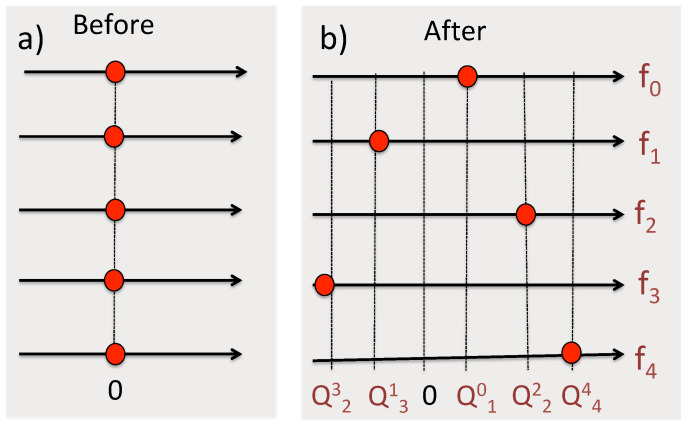
The measurements shown in Figure 3, this time made by employing five accurate von Neumann pointers. (**a**) Initially, the pointers are set to zero. (**b**) At the end of the trial, one finds each pointer shifted by the corresponding eigenvalue. As in Figure 3, an outcome $\left\{Q_{4}^{4}\leftarrow Q_{2}^{3}\leftarrow Q_{2}^{2}\leftarrow Q_{3}^{1}\leftarrow Q_{1}^{0}\right\}$ is recorded.

**Figure 5 entropy-24-00877-f005:**
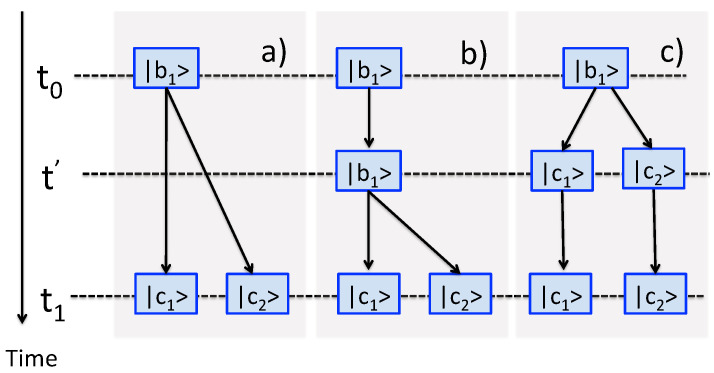
(**a**) A measurement of an operator $\hat{B}=\sum_{i=1}^{2}\left|b_{i}\rangle B_{i}\langle b_{i}\right|$ prepares the system (${\hat{H}}_{S}=0$) in a state $|b_{1}\rangle$, and is followed by a measurement of $\hat{C}=\sum_{i=1}^{2}\left|c_{i}\rangle C_{i}\langle c_{i}\right|$. (**b**) An additional measurement of $\hat{B}$ at $t_{0}<t^{\prime }<t_{1}$ yields an outcome $B_{1}$, and finds the system in the state $|b_{1}\rangle$ with certainty. (**c**) An additional measurement of $\hat{C}$ at the same $t^{\prime }$ yields $C_{i}$ with certainty, if the last outcome is also $C_{i}$. The probabilities are unchanged, $P\left(C_{i}\leftarrow B_{1}\right)=P\left(C_{i}\leftarrow B_{1}\leftarrow B_{1}\right)=P\left(C_{i}\leftarrow C_{i}\leftarrow B_{1}\right)$, and it would appear that at $t=t^{\prime }$ the system has well defined values of non-commuting operators $\hat{B}$ and $\hat{C}$.

**Figure 6 entropy-24-00877-f006:**
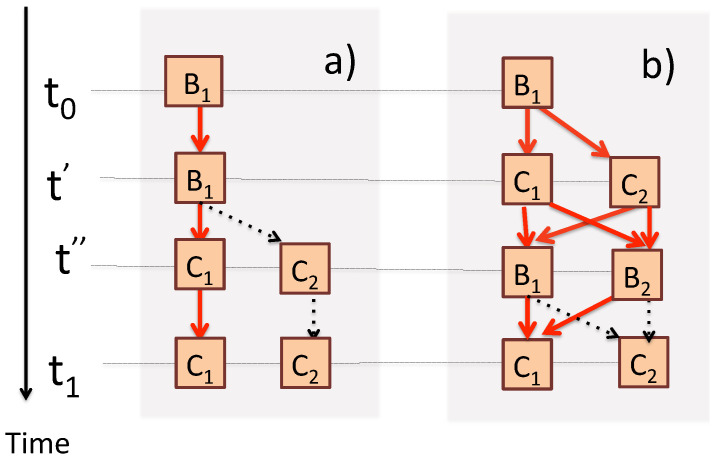
(**a**) In the case shown in Figure 5b, at $t^{\prime }<t^{\prime \prime }<t_{1}$ one can add a measurement of $\hat{C}$, still leaving the probabilities unchanged, $P\left(C_{i}\leftarrow B_{1}\right)=P\left(C_{i}\leftarrow C_{i}\leftarrow B_{1}\leftarrow B_{1}\right)$. It would appear that in a transition $\left\{C_{1}\leftarrow B_{1}\right\}$ intermediate values $B_{1}$ and $C_{1}$ co-exist for any $t^{\prime }$ and $t^{\prime \prime }$, such that $t^{\prime \prime }>t^{\prime }$. (**b**) The above is no longer true if $\hat{C}$ is measured before $\hat{B}$. The transition between the cases (**a**,**b**) is discussed in Section 7.

**Figure 7 entropy-24-00877-f007:**
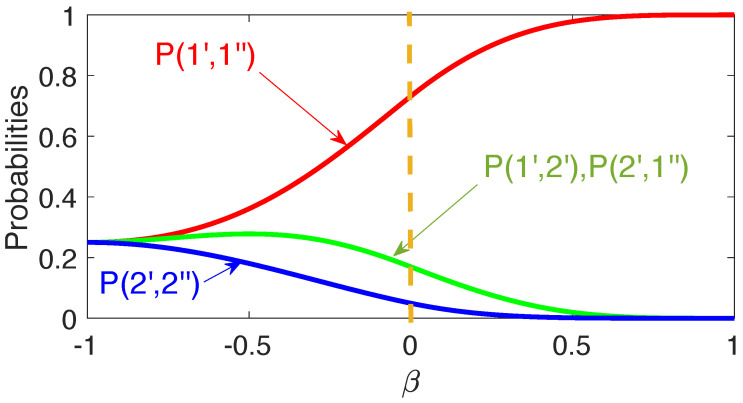
The values of $\hat{B}={\hat{\sigma }}_{x}$ and $\hat{C}={\hat{\sigma }}_{y}$ are measured jointly (cf. Equations (Equation 36)–(Equation 38)) for a two-level system (${\hat{H}}_{S}=0$), initially polarised along the *x*-axis, $|b_{1}\rangle ={|↑}_{x}\rangle$, $B_{1}=1$, and later found polarised along the *y*-axis, $|c_{1}\rangle ={|↑}_{y}\rangle$, $C_{1}=1$. Four probabilities in Equation (Equation 40) are plotted vs. β in Equation (Equation 35). For β=1 the measurement of $\hat{B}$ precedes that of $\hat{C}$, for β=0, $\hat{B}$ and $\hat{C}$ are measured simultaneously, and for β=−1, $\hat{C}$ is measured first.

**Figure 8 entropy-24-00877-f008:**
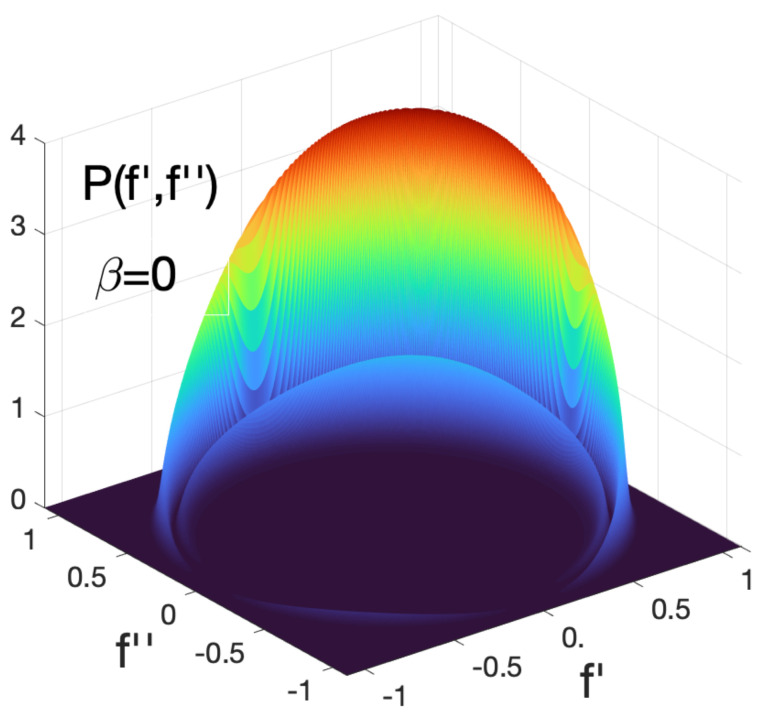
The values of $\hat{B}={\hat{\sigma }}_{x}$ and $\hat{C}={\hat{\sigma }}_{y}$ are measured simultaneously (β=0, Δf=0.05) for a two-level system (${\hat{H}}_{S}=0$), initially polarised along the *x*-axis, $|b_{1}\rangle ={|↑}_{x}\rangle$, $B_{1}=1$, and later found polarised along the *y*-axis, $|c_{1}\rangle ={|↑}_{y}\rangle$, $C_{1}=1$.

**Figure 9 entropy-24-00877-f009:**
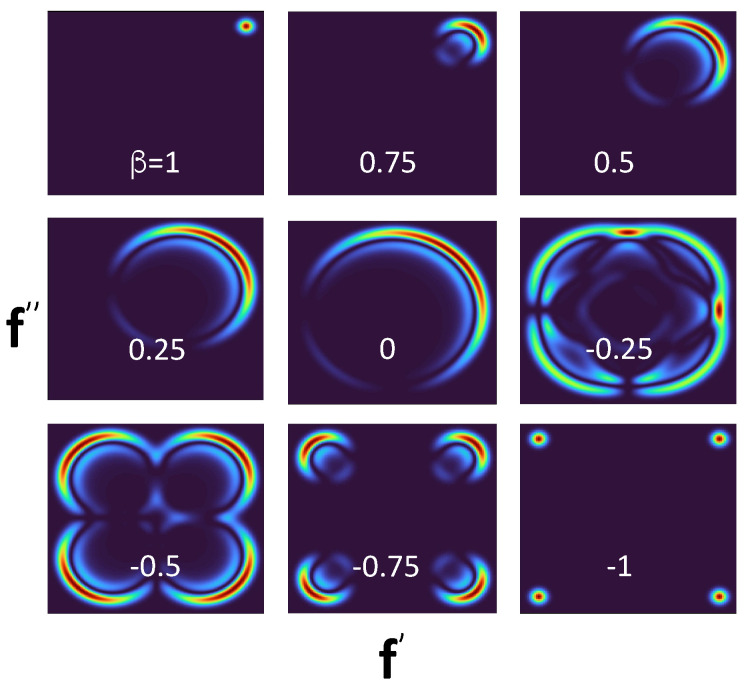
The same probability as in Figure 8, but for different degrees of overlap between the measurements. The value of β in Equation (Equation 34) is indicated in each panel. The cases β=1 and β=−1 correspond to the scenarios (a) and (b) shown in Figure 4, respectively.

## Data Availability

Not applicable.

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
