# Peer review of "Unitary Evolution and Elements of Reality in Consecutive Quantum Measurements"

_entropy, 2022, doi:10.3390/e24070877_

Round 1
Reviewer 1 Report
This is a good paper that addresses consecutive measurements from an orthodox (Feynman) perspective, in a minimalist view of the formalism that does not give too much physical content to the wavefunction and provides interesting analysis of physical reality, or the role of the observations. The conclusions are correct as far as I can tell, and not trivial to find. The analysis is very well suited to the Special Issue. The author is a recognized expert in foundation of quantum mechanics. I find the references adequate as well as the presentation.
Author Response
I can only thank the Reviewer for his/her report.
The spell check has been carried out.
Reviewer 2 Report
I think this is an interesting paper and recommend it for publication. I found the following typos that should be corrected.
page line correction
2 8 thought experiments
2 8 we have
3 23 In equation (1) the middle expression is missing the
eigenvalue term.
5 16 one will not even be able
8 -5 that it does
8 -1 and retain
11 7 continues on
14 2 goes as follows
16 -4 omit }
16 -3 C^ at t
22 11 discrete
26 3 records of the
26 -6 but not forgetting
27 -6 types of probes
Author Response
I am grateful to the Reviewer, and can only apologise for not weeding out
the typos earlier. They are corrected now:
page line correction
2 8 thought experiments. -corrected, thank you
2 8 we have -corrected, thank you
3 23 In equation (1) the middle expression is missing the
eigenvalue term- so it does, thank you
5 16 one will not even be able -corrected, thank you
8 -5 that it does -corrected, thank you
8 -1 and retain -corrected, thank you
11 7 continues on -corrected, thank you
14 2 goes as follows -corrected, thank you
16 -4 omit } -corrected, thank you
16 -3 C^ at t -corrected, thank you
22 11 discrete -corrected, thank you
26 3 records of the -corrected, thank you
26 -6 but not forgetting -corrected, thank you
27 -6 types of probes - I would rather leave it as it is.
The idea is that the result of a calculation which involves the probes can be obtained by manipulating the system's amplitudes. One shouldn't, however, forget that the probes are responsible for the breaks in the system's unitary evolution.